

# Tsunami risk perception in Southern Italy: first evidence from a sample survey.

Andrea Cerase[1,2], Massimo Crescimbene[1], Federica La Longa[1] and Alessandro Amato[1],

[1] Istituto Nazionale di Geofisica e Vulcanologia, Roma, 00143, Italy
[2] Department of Communication and Social Research, La Sapienza University, 00198, Roma Italy and Istituto Nazionale di Geofisica e Vulcanologia, Roma, 00143, Italy

*Correspondence to*: Andrea Cerase (andrea.cerase@gmail.com)

**Abstract.** According to a deep-rooted conviction, the occurrence of a tsunami in the Mediterranean Sea would be very rare. However, in addition to the catastrophic event of Messina and Reggio Calabria (1908) and the saved danger for the tsunami occurred on Cycladic sea in 1956, 44 events are reported in the Mediterranean Sea between 1951 and 2003, and other smaller tsunamis occurred off Morocco, Aegean and Ionian seashores between 2017 and 2018. Such events, that are just a little part of the over 200 historically events reported for the Mediterranean (Maramai, Brizuela & Graziani, 2014) should remind geoscientists, civil protection officers, media and citizens that 1) tsunami hazard in the Mediterranean is not negligible, and 2) tsunamis come in all shapes and colours, and even a small event can result in serious damages and loss of lives and properties. Recently, a project funded by the European Commission (TSUMAPS-NEAM, Basili et al., 2018) has estimated the tsunami hazard due to seismic sources in the NEAM region (one of the four ICG coordinated by the UNESCO IOC) finding that a significant hazard is present in most coasts of the area, particularly in those of Greece and Italy. In such a scenario, where low probability and high uncertainty match with poor knowledge and familiarity with tsunami hazard, risk mitigation strategies and risk communicators should avoid undue assumptions about public's supposed attitudes and preparedness, as these may results in serious consequences for the exposed population, geoscientists, and civil protection officers. Hence, scientists must carefully shape their messages and rely on well-researched principled practices rather than on good intuitions (Bostrom, & Löfstedt, 2003).

For these reasons, the Centro Allerta Tsunami of the Istituto Nazionale di Geofisica e Vulcanologia (hereinafter CAT-INGV) promoted a survey to investigate tsunami's risk perception in two pilot regions of Southern Italy, Calabria and Apulia, providing a stratified sample of 1021 interviewees representing about 3.2mln people living in 183 coastal municipalities of two regions subjected (along with Sicily) to relatively high probability to be hit by a tsunami. Results show that people's perception and understanding of tsunami are affected by media accounts of large tsunamis of 2004 (Sumatra) and 2011 (Tohoku, North East Japan): television emerged as the most relevant source of knowledge for almost 90% of the sample, and the influence of media also results in the way tsunami risk is characterized. Risk perception appears to be low: for almost half of the sample the occurrence of a tsunami in the Mediterranean sea is considered quite unlikely. Furthermore, the survey's results show that the word 'tsunami' occupies a different semantic space with respect to the Italian traditional headword 'maremoto', with differences among sample strata. In other words, the same physical phenomenon would be understood in two different ways by younger, educated people and elders with low education level. Also belonging to different coastal areas[1] appears to have a significant influence on the way tsunami hazard is conceived, having a stronger effect on risk

---

[1] For the purposes of this paper, the term "coastal area" refers to the part of the coastline defined by both seas and regions' limits, according to current geographical conventions. The Tyrrhenian Calabria indicates the coastline between coastal municipalities of Tortora and Scilla; Ionian Calabria spans from Reggio Calabria to Rocca Imperiale; Ionian Apulia from Ginosa to Castrignano del Capo and Adriatic Apulia from Gagliano del Capo and Chieuti.



characterization, for instance the interviewees of Tyrrhenian Calabria are more likely to associate tsunami risk to volcanoes
with respect to other considered coastlines. The results of this study provide a relevant account of the issues at a stake, also
entailing important implication both for risk communication and mitigation policies.


**1. Introduction**
**1.1 Relevance of tsunami risk in the Mediterranean and Italian coasts**
Almost all countries surrounding the Mediterranean have faced the effects of historical tsunamis in the past four millennia,
with more than 200 events documented for the area, as shown in the catalogue published by Maramai et al. (2014). According
to this catalogue, most of the tsunamis in the area (~83%) have been generated by earthquakes, a fraction similar to that of
other oceanic regions worldwide. Since 1700 AD, an average of 20 events every 50 years (including small ones) is reported in
the catalogue (Maramai et al., 2014), i.e., one event every 2.5 years.
Besides large historical tsunamis, such as the big one occurred for an earthquake in Crete in 365 DC (cit.), in the 20th century
at least two important events did occur: The 1908 tsunami in southern Italy (Messina, Reggio Calabria and the surrounding
coasts in 1908) due to a magnitude 7 earthquake in the Messina Straits, with run-up as high as 13m in Pellaro (Tinti and
Maramai, 1996) the 1956, magnitude 7.7 earthquake that occurred close to the Cycladic island of Amorgos (Greece)
triggering large tsunami waves hitting the coasts of Amorgos, Astypalaia and Folegandros with run-up values of 20, 10, and
14 m, respectively (Okal et al., 2009), up to 30m according other sources (Ambraseys, 1960). More recently, in 2003 a
relatively small tsunami caused by a magnitude 6.9 earthquake in Boumerdes (Algeria) hit the Western Mediterranean coasts
causing damage properties in at least eight harbours in Balearic Islands (Vela et al. 2011). Finally, two small tsunamis
occurred in Dodecanese in 2017 (magnitude 6.4 and 6.6), along with the most recent one occurred in Ionian Sea (Zakynthos)
in October 2018 (magnitude 6.8).
Based on these and other geological data, the first probabilistic hazard assessment for tsunamis (of seismic origin) (S-PTHA)
in the NEAM region has been computed and published (TSUMAPS-NEAM Team, 2018). In an S-PTHA approach, the hazard
in any specific point on the coast comes from the various tsunami sources affecting that point, including close and distant
sources (Selva et al. 2016; Grezio et al, 2017; Davies et al., 2017; Volpe et al. 2019). For Italy, it is evident that the most
hazardous areas are those exposed to both local earthquakes and distant ones. In particular, the most active region in the
Mediterranean is the Hellenic arc, where strong tsunamigenic earthquakes have occurred in the past (Papadopulos et al., 2010;
Maramai et al., 2014). Consequently, the coastal areas of Apulia, Calabria and Eastern Sicily facing the Ionian sea, have the
highest hazard in Italy (Italian Civil Protection, 2018).
However, a significant hazard exists for many other coastal areas throughout Italy, as the Ligurian Sea, the Adriatic Sea, and
also the Tyrrhenian Sea, due to either local earthquake sources or distant ones, as for instance the northern African fault
system from Gibraltar to Tunisia.
Despite the high hazard of the Italian coasts, the memory of tsunamis is weak in Italy, mainly due to the long time elapsed
since the last deadly event in 1908. In that circumstance the tsunami increased significantly the already heavy death toll by the
earthquake, also due to the unawareness of people about the tsunami risk: Many people escaped from the damaged and
dangerous streets of Messina and other towns, looking for a safe place near the sea. After more than one century from this
tragedy, we do not know if some memory has left in the region.
Another recent event that could have modified the perception of tsunami risk in Italy is the collapse of the unstable flank of
the volcanic island of Stromboli in 2002, that generated a local tsunami, with measured run-up up to 10 m (Tinti et al., 2005).



## 1.2 The general tsunami context in the Mediterranean and the CAT-INGV (mission, national and international role)

Coastal areas bordering the Mediterranean basin are subject to tsunami hazard. For this reason, in 2005 the Intergovernmental Oceanographic Commission of UNESCO (IOC-UNESCO) established the Intergovernmental Coordination Group for the Tsunami Early Warning and Mitigation System in the North-eastern Atlantic, the Mediterranean and connected seas (ICG/NEAMTWS), in response to the tragic 'Boxing Day' tsunami of December 26[th] 2004, in which over 230,000 lives were lost around the Indian Ocean region. Nowadays the Mediterranean coasts are one of the most densely populated areas of the world, with about 130 million people living along a 46.000 km coastline, 230 million tourists visiting the Mediterranean Sea venues every year, and 7 coastal cities with more than 2 million inhabitants (Marriner et al. 2017). Mediterranean Sea also fosters a thriving maritime economy: according to the estimates of WWF-BCG report economic activities related to the Mediterranean worth US$ 450 billion for year (Randone et al. 2017). Hence, the increasing anthropization of the Mediterranean coasts, along with the enhanced relevance of tourism-related activities, make it particularly important to improve risk mitigation strategies in the area.

Following the NEAMTWS establishment, Italy has started to build a tsunami alert centre at the Istituto Nazionale di Geofisica e Vulcanologia (INGV) in 2013. After a three-year testing phase, the CAT-INGV has become operational in 2016, after the accreditation by the ICG/NEAMTWS as a Tsunami Service Provider for the whole Mediterranean area. Soon after that, the CAT-INGV became operational at national level within the framework of the so called SiAM (Sistema d'Allertamento nazionale per i Maremoti di origine sismica), coordinated by the Italian Department of national Civil Protection, a Prime Minister Office, and together with the Istituto Superiore per la Protezione dell'Ambiente (ISPRA), which manages the national sea level network.

As a Tsunami Service Provider, CAT-INGV sends alert messages to about fifteen countries and Institutions of the Euro-Mediterranean region in case of potentially tsunamigenic earthquakes. At national level, CAT-INGV cooperates strictly with DPC and ISPRA for disseminating alert messages to the local authorities and the population. As well, CAT-INGV is involved in increasing knowledge and people awareness on the tsunami hazard and risk.

## 2. Why a research?

Tsunami risk mitigation strategies might definitely benefit from risk perception research, also contributing to enhance people's ability to understand phenomena and to enforce both individuals' and communities' response capabilities. Comprehensive and sound risk communication strategies should rely on well-researched principles rather than unproven assumptions about people's attitudes toward risk (Bostrom & Löfstedt, 2003).

The availability of robust data on tsunami risk perception may thoroughly improve the effectiveness of mitigation measures, hence, the decision to implement and test a replicable and extensible research model to be applied first in the pilot regions and then elsewhere. This pilot study has three strategic goals: 1) to provide empirical data on citizens' understanding and risk perception in a tsunami risk prone area, also allowing future comparisons with different areas of the NEAM Region; 2) to identify the most appropriate key messages, channels, and techniques to effectively communicate risk in peacetime as a necessary precondition of effective early warning in case of an event; 3) to enable / improve scientific communication strategies and activities to be implemented by the Italian Tsunami Alert Centre CAT–INGV as a part of its mandate, including the development of a dedicated website and social media channels.

Such research is also intended to provide a first basis for a national and cross-national comparison of survey results, as to get a comprehensive picture of prior knowledge about tsunamis and risk perception among residents of different regions and countries, exploring both common traits and distinctive characteristics (Kurita et al., 2007)



### 3. Risk perception

The perception of risks involves the process of collecting, selecting and interpreting signals about uncertain impacts of events, activities or technologies. These signals can refer to direct observation or information from others (for example reading about an earthquake in the newspaper). Perceptions may differ depending on the type of risk, the risk context, the personality of the individual, and the social context.

Within natural sciences the term 'risk' seems to be clearly defined, it means the probability distribution of adverse effects, but the everyday use of the word 'risk' has different connotations (Renn, 2008). For social sciences the terminology of 'risk perception' has become the conventional standard (Slovic, 1987). Yet risks cannot be 'perceived' in the sense of being taken up by the human senses, as are images of real phenomena. The mental models and other psychological mechanisms through which people use to judge risks (such as cognitive heuristics and risk images) are internalized through social and cultural learning and constantly moderated (reinforced, modified, amplified or attenuated) by media reports, peer influences and other communication processes (Morgan et al., 2001).

### 3.1 Theoretical references of studies on risk perception

In recent decades, many research studies have been carried out on psychological, social and cultural factors that influence the perception of risk. At present, the perception of risk is considered fundamental to understand what lay people think about risk and to adopt suitable political and communication strategies to cope with it.

Renn and Rohrmann (2000) developed a structured framework that provides an integrative and systematic perspective on risk perception. Figure 3.1 illustrates this perspective by suggesting four distinct context levels (originally presented by Renn and Rohrmann, 2000: 221; adapted from Breakwell's (1994) generic model.

The first level includes the collective and individual heuristics that individuals apply during the process of forming judgements. These heuristics are independent of particular risk nature, personal beliefs, emotions or other conscious perception patterns of the individual. Heuristics represent common-sense reasoning strategies that have evolved over the course of biological and cultural evolution (Ross 1977; Kahneman and Tversky, 1979; Breakwell, 2007). They may differ between cultures; but most evidence from psychological research shows a surprising degree of universality in applying these heuristics across different cultures (Renn and Rohrmann, 2000).

The second level refers to the cognitive (knowledge-based) and affective (emotion-based) factors that influence the perception of specific properties of the risk in question. Cognition about a risk source – what people believe to be true about a risk – governs the attribution of qualitative characteristics (psychometric variables) to specific risks (e.g. dread or personal control options) and determines the effectiveness of these qualitative risk characteristics on the perceived seriousness of risk and the judgement about its acceptability (Slovic, 1992). Recently, psychologists have discovered that affect and emotions play an important role in people's decision processes (Loewenstein et al, 2001; Slovic et al, 2002). People's feelings about what is good or bad in terms of the causes and consequences of risks colour their beliefs about the risk and, in addition, influence their process of balancing potential benefits and risks.

The third level refers to the social and political institutions that individuals and groups associate with either the cause of the risk or the risk itself. Most studies on this level focus on trust in institutions, personal and social value commitments, organizational constraints, social and political structures, and socio-economic status. One important factor in evaluating risk is the perception of fairness and justice in allocating benefits and risks to different individuals and social groups (Linnerooth-Bayer and Fitzgerald, 1996).


Other studies have placed political and social organizations, and their strategies to communicate with other organizations and
society at large, as the prime focus of their attention (Clarke, 1989; Shubik, 1991). Press coverage appears to contribute
substantially to a person's perception of risk, particularly if the person lacks personal experience with the risk and is unable to
verify claims of risks or benefits from their own experience. In contrast to popular belief, however, there is no evidence that
the media create opinions about risks or even determine risk perceptions. Studies on media reception rather suggest that
people select elements from media reports and use their own frame of reference to create understanding and meaning. Most
people reconfirm existing attitudes when reading or viewing media reports (Peters, 1991; Dunwoody & Peters, 1992;
Breakwell 2007).
The last level refers to cultural factors that govern or co-determine many of the lower levels of influence. The most specific
explanation for cultural differences about risk perceptions comes from the so-called 'cultural theory of risk'.
**Fig. 3.1 Levels of Risk Perception (Renn, 2008)**

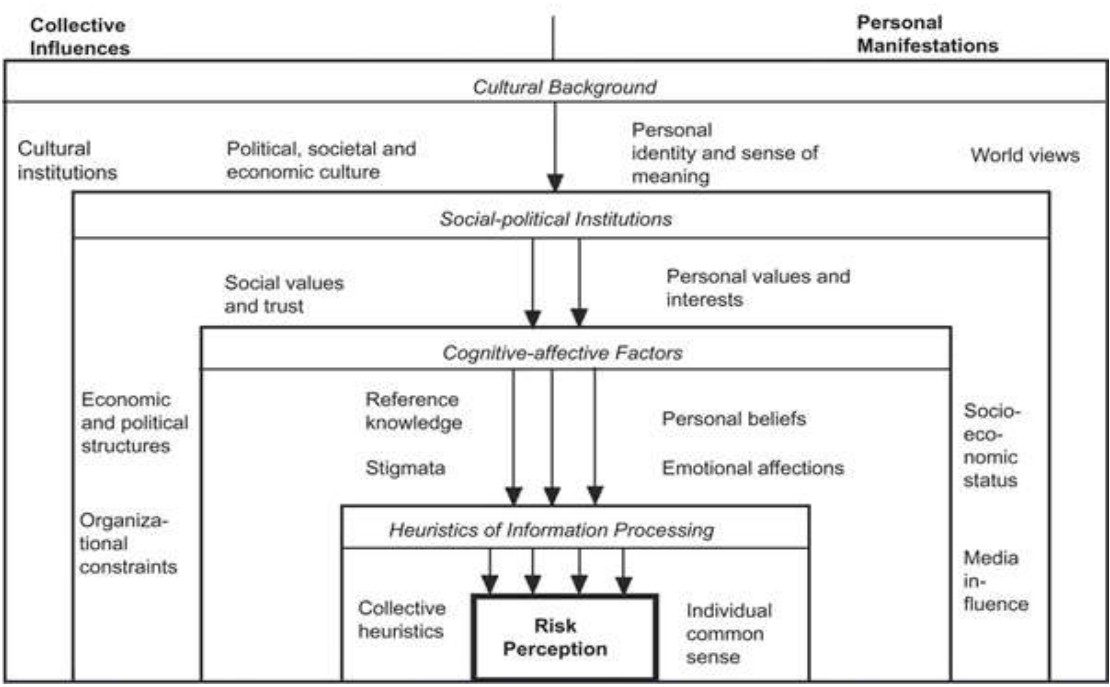

Cultural theory claims that there are four (five according some authors) prototypes of responses to risk (Thompson, 1980;
Douglas and Wildavsky, 1983; Thompson et al, 1990). These ideal-types refer to entrepreneurs, egalitarians, hyerarchists,
atomized individuals and, as the fifth separate category, hermits. Opinions on the validity of the cultural theory of risk differ
widely. All authors agree, however, that specific culture-based preferences and biases are, indeed, important factors in risk
perception. The disagreement is about the relevance of the postulated four or five prototypes within the realm of cultural
factors. In addition to the theory of cultural prototypes, there are two sociological concepts that provide plausible explanations
for the link between macro-sociological developments and risk perceptions. The theory of reflexive modernization claims that
individualization, pluralisation and globalization have contributed to the decline of legitimacy with respect to risk



professionals and managers (Beck, 1994; Mythen, 2005). Due to this loss of confidence in private and public institutions,
people have become skeptical about the promises of modernity and evaluate the acceptability of risks according to the
perceived interest and hidden agenda of those who want society to accept these risks (Beck, 1992). The second approach picks
up the concept of social arenas in which powerful groups struggle for resources in order to pursue their interest and objectives.
Here, symbolic connotations constructed by these interest groups act as powerful shaping instruments for eliciting new beliefs
or emotions about the risk or the source of risk (Renn, 1992; Jaeger et al, 2001).
All four levels of influence are relevant in order to gain a better and more accurate understanding of risk perception. In spite
of many questions and ambiguities in risk perception research, one conclusion is beyond any doubt: abstracting the risk
concept to a rigid formula, and reducing it to the two components' 'probability and consequences', does not match people's
intuitive thinking of what is important when making judgements about the acceptability of risks (Slovic, 1992). The
framework of social amplification may assist researchers and risk managers to forge such an integrative perspective on risk
perception. Yet, a theory of risk perception that offers an integrative, as well as empirically valid, approach to understanding
and explaining risk perception is still missing (Wachinger & Renn, 2010).
**3.2      Recent research on tsunami risk perception**
Unfortunately, research on tsunami risk perception in the Mediterranean are neither numerous nor homogeneous, and it seems
to be lacking a sufficiently broad and coherent framework. Nevertheless, one of the oft-cited figures in research is a tendency
of coastal populations to underestimate tsunami risk, assumed being quite negligible. Such an issue tends to manifest itself
under various forms and levels in different countries, both those with a long story of tsunamis - even in recent times - and
those hit by events dating back centuries, a long time after social memory faded away. Furthermore, both quantitative and
qualitative research on tsunami risk perception highlights that the idea of *subjective immunity* (Douglas, 1986) can be
motivated by a number of different factors, related to both psychological and cultural issues. Such factors include risk denial,
lack of experience with similar events, unrealistic optimistic bias, poor understanding of tsunami dynamics and of critical
height of the waves, lessened urgency to adopt countermeasures after strong earthquake with no tsunami also considering the
way religious beliefs may result into a fatalistic approach toward natural hazards (Oki & Nakayachi, 2012, Couling, 2014;
Setiadi, 2016; Alam, 2016; Paton et al., 2017).
Other research has been previously carried out in the NEAM area. In 2014 the EU – funded project ASTARTE investigated
tsunami risk perception and community preparedness, for an overall total of 1159 questionnaires achieved in six seaside
venues in just as many countries (France, Greece, Norway, Portugal, Spain, Turkey). The survey was based on a standardized
questionnaire (about 50 questions), and random face-to-face interviews being administered on main beaches, boats, ports, city
centers (Papageorgiou et al., 2015; Goeldner-Gianella et al., 2017; Liotard et al., 2017). Despite the precious insights coming
from this investigation, research design was a critical issue. Unfortunately, random interviews together with small size
samples pose serious issues as regards methodology, since results' reliability is a concern. Sample size is a very relevant issue
to be addressed in such kind of research, as statistical significance is the precondition to draw any solid conclusion from
questionnaire surveys (Raine, 1995; Bird & Dominey-Howes, 2008). Although face-to-face random interviews are less costly
and time consuming, researchers have definitely no way to control the profile of respondents in order to check if interviewees
composition would fit or not with the demographic profile of the considered population, and even less to ground any sound
risk mitigation strategy. Limitations might be carefully addressed, and such a method should be cautiously deployed in the
early stage of a research, for instance to identify or test major issues and topics.
The 2004 Boxing Day event and its global scale consequences revealed how cultural and societal resources have actually
resulted into different abilities to cope with tsunami risk, thus triggering a fresh new interest for risk perception research in its



broadest sense, also including psychological, sociological and anthropological approaches. The lack of information or of
cultural memory of past events, including their negative outcomes, may jeopardize the effectiveness of any mitigation
program, while the improvement of knowledge and emergency plans should be prioritized. Such programs should not be
handled down from on high, but must be always placed within a given social context. Involved communities should indeed
mediate between agencies' proposals and pre-existing knowledge through a variety of patterns of relationship, which should
always include and properly consider the value of participation, self-efficacy, empowerment and trust (Paton et al., 2008).

**3.3. Research hypothesis**

Our research lies on a general assumption: The lack of awareness and the misconceptions about tsunami dynamics and impact
may considerably hamper the effectiveness of mitigation measures. As a consequence, the effectiveness of risk
communication and community engagement strategies should rely upon a clear-cut definition of the issues to be fixed. More
in detail, the scope of this paper is to provide a first verification of the two following hypothesis:
- RH1: the way tsunami risk is characterized depends on people's sources of knowledge and their ability to affect risk
perception. Such characterization, and people's expectancies about tsunami significantly depends on media
representations of catastrophic events such as those occurred in Sumatra and Japan.
- RH 2: Risk perception is influenced by socio-demographic variables such as age, gender and education, which can
result in different beliefs about tsunami and its related phenomena.

**3.4. Methods and techniques (questionnaire)**

According to a well-established standard in social science and risk perception research, questionnaire survey was deemed to
be the most suitable method of investigation, in line with research general goals. The need to construct a reliable database to
get an insight into public's awareness of tsunamis and related risk perception/understanding, along with the need to support
analysis with statistical evidence and to guarantee full comparability for further research, have led to the decision of using
such a methodology as a starting point of a wider research strategy. Data collection has been operated by Questlab S.r.l., a
specialized research company based in Venice, strictly following research team directions about a) reference universe, b)
sampling strategy, c) stratification variables, d) number of interviews to be implemented and administered. Interviews have
been carried out by using Computer Aided Telephonic Interview (CATI) methodology. The research covered two regions of
Southern Italy, Calabria and Apulia, as to represent over three million inhabitants living in some of the most tsunami prone
areas of the Italian peninsula, as it appears from historical catalogues of the Italian tsunamis (Tinti et al, 2004; Maramai et al.,
2013) and S-PTHA studies (Lorito et al. 2008; Basili et al., 2013).
Research was carried out on a proportional stratified sample of 1,021 respondents, including 474 men and 547 women aged
18-95 years across 138 different coastal municipalities of Apulia and Calabria. It is worth recalling that Apulia and Calabria
shorelines have an extension of respectively 865 km and 780 km long, covering 22% of Italian coasts and 16% of the whole
population of Italian population residing in coastal municipalities.
The sampling plan was aimed at ensuring the best possible statistical representativeness with the available resources, in order
to provide scientists, end-users and Civil Protection with robust and reliable data to ground both mitigation actions, also
improving scientific debate on these topics. Interviewees were selected by using three stratification variables: age, gender and
coastal areas, as to guarantee the best possible correspondence between subpopulations in the sample and in the reference
universe. 833 questionnaires were administered to landlines users and other 188 to mobile phone users, for a total of 1,021
questionnaires. The decision to contact mobile phones users was due to the need to involve a larger number of young people





and males, who are less likely to use landlines instead of mobile phones (Censis, 2018)[2]. Data collection was completed
between April and May 2018 by a team of over twenty trained interviewers, supervised by highly trained research experts.

**Tab. 1 - Sample of the survey for age, gender, regional coast and educational level**

| Coastal area / Education level / Age | Ionian Calabria | | | Tyrrhenian Calabria | | | Adriatic Apulia | | | Ionian Apulia | | | Total |
|---|---|---|---|---|---|---|---|---|---|---|---|---|---|
| | L | I | H | L | I | H | L | I | H | L | I | H | |
| 18-49 | 1 | 44 | 49 | 1 | 31 | 22 | 3 | 96 | 64 | 0 | 31 | 17 | 359 |
| 50-64 | 5 | 58 | 33 | 2 | 29 | 10 | 4 | 106 | 28 | 2 | 41 | 13 | 331 |
| over 64 | 13 | 46 | 23 | 6 | 20 | 11 | 23 | 83 | 41 | 7 | 46 | 12 | 331 |
| Total | 19 | 148 | 105 | 9 | 80 | 43 | 30 | 285 | 133 | 9 | 118 | 42 | 1021 |

L = Low level of education or no instruction; I = Intermediate, secondary school and high school; H = graduate and post-graduate

**Tab. 2a – Response rate for age and channel**

| | Landline | Mobile | Total |
|---|---|---|---|
| 18-34 yrs. | 11,0% | 29,8% | 14,5% |
| 35-49 yes | 14,0% | 50,0% | 20,7% |
| 50-64 yes | 35,7% | 18,1% | 32,4% |
| 65 and over | 39,3% | 2,1% | 32,4% |
| Total | 100,0% | 100,0% | 100,0% |


**Tab. 2b – Response rate for gender and channel**

| | Landline | Mobile | Total |
|---|---|---|---|
| Men | 42,9% | 62,2% | 46,4% |
| Women | 57,1% | 37,8% | 53,6% |
| Total | 100,0% | 100,0% | 100,0% |



---

[2] Sampling operations followed these steps: (a) defining the population; (b) choosing sample size; (c) listing the population; (d) assigning numbers to cases; (e) calculating the sampling fraction; (f) selecting the first unit; and (g) selecting our sample.




**Fig. 1: geographical distribution of interviewees' municipalities**

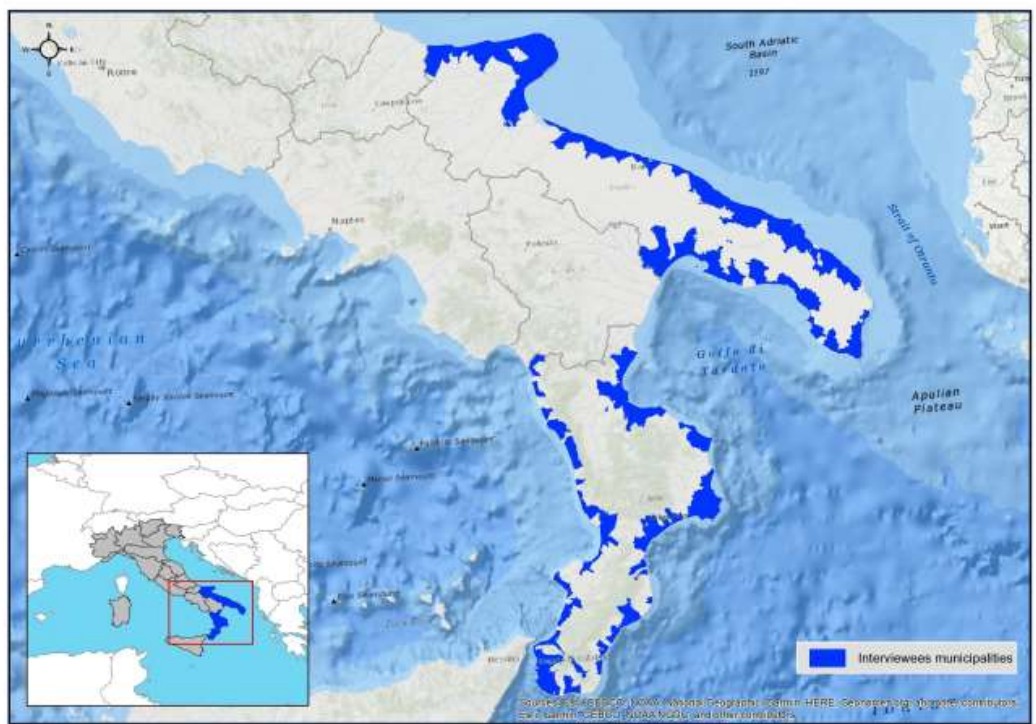


**4. Discussion**
**4.1 Tsunami: sources of knowledge and risk characterization**
According to hypothesis 1, the ways tsunami and related risk are perceived and understood are affected by the sources of
knowledge which have been actually used by the interviewees, first and foremost the media. More specifically, the way risk is
characterized by common people strictly depends on the sources which are actually available to people and their ability to
handle such an information, resulting into a variety of mental models of phenomena and of their possible consequences. In
this paper, the concept of risk characterization is to be intended as the way lay-people identify relevant attributes of a certain
hazard and rate their importance as a base for their individual risk assessment (Fischhoff & Morgan, 2013), rather than
referring to a formalized expert-judgement process to estimate probability, magnitude and potential harm. Although both have
similar characteristics, authors exclusively refer to the social process by which common people recall as relevant certain risk
attributes instead of others.
We first considered the difference between the Japanese word 'tsunami' which dominates the tsunami risk governance field
and the word 'maremoto' (literally sea-quake), that is more common in spoken Italian. Survey results have shown that these
two words are associated with two different mental models, in which some given features of the phenomenon are differently
recalled and combined together, although with some degree of overlapping. Putting aside minor differences, the idea of 'big
wave' is strongly associated with the word 'tsunami' (60,8 %) rather than with 'maremoto' (39.5%). Moreover, the word
earthquake ('terremoto' in Italian) is mentioned as a feature of 'maremoto' (50%) more frequently than with 'tsunami'



(35.4%). Other differences are found for the association of 'maremoto' with sea-storms (23,7% vs. 17,9% for 'tsunami'),
while sea withdrawal is slightly more associated to tsunami (15,9%) than to 'maremoto' (10,9%). In general terms, the
majority of the interviewees considers more familiar the Italian word 'maremoto' (53,3%) than 'tsunami' (46,7%). Such a
difference is more pronounced for elders, for women, and for people with low level of education, and of course has relevant
implication for future risk communication strategies. It would be interesting to verify whether similar differences are present
in other languages, where 'autochthonous' words such as the Italian 'maremoto' do exist, and the same word is also used in
Spanish.
Interviewees were asked to respond about the possible causes of tsunamis: earthquakes are correctly recalled by 75% of
respondents, while volcanic eruptions were indicated by 46,1%, meteorological phenomena by 12,2%, meteorites befalling in
the sea by 10,1%, landslides by 9.0%, and finally 6% proposed other possible causes[3]. Bivariate analysis has shown that listed
causes are first influenced by coastal area, and then by level of education, age and gender: further analyses are required to
better explain these differences. Such percentages reflect in some way the relative distributions of tsunamis' causes
worldwide. Although the interviewees had the possibility to select more than one choice (and therefore they could have
selected all of them), it is possible that they decided to pick only a few of them, i.e., those that were thought as more likely.
As previous data suggest, people are more likely to recall some aspects of the physical phenomenon instead of others, which
appear to be less familiar. Data distribution simply shows that the first five items, arranged in decreasing number of 'correct'
answers, are absolutely consistent with the catastrophic visual imagery of the great tsunamis of Sumatra and Tohoku, as most
of interviewees were able to address physical damages to houses, building and infrastructures (92,2%); negative impacts on
economy and occupation (91,6%), environment (90,4%); casualties and injured people (89,4%). An interesting result
emerging from the survey is that people are well aware that fleeing to the beach after a strong shake is not the right choice
305 (85,1%).

The greatest difficulties to understand tsunamis are concerned with some relatively unfamiliar effects, such as the possibility
to have great tsunamis (> 20m) even in the Mediterranean (38,6%); that tsunami may trigger strong sea currents (37,8%) and
that a tsunami wave of only 50cm can be actually dangerous for people staying near the shorelines (19,2%). Evidence from
this survey are consistent with what happened in the aftermath of recent events occurred in the Mediterranean. On July 21st
2017, a small tsunami stroke the island of Kos (Greece) and the nearby coastal city of Bodrum (Turkey) with run-up elevation
up as high as 2m (Yalçiner et al., 2017). On that occasion, surveillance cameras on Kos waterfront captured the way people
were reacting to sea - level anomalies: they were seemingly calm and utterly curious to see the water inundating quaysides.
They were shooting pictures and videos with their smartphones instead of fleeing away, thus emphasizing that the risk posed
by small tsunamis was almost completely ignored.
In order to get a concise and comprehensive picture of knowledge about the phenomenon, a rough but effective knowledge
index has been developed, simply calculated as the unweighted sum of the number of correct answers to the whole above
listed questions about the 'physical reality' of the event divided by the number of items considered. Given the average value
for the whole sample (0.6952), gender, level of education, age and coastal area differently affect the level of knowledge. It is
evident that a higher level of education implies a higher index of knowledge. Index's value is anyway higher (> average
value) for women, for middle-aged people (35-49), for residents of Tyrrhenian Calabria and Ionian Apulia coastal areas. On
the contrary, elder people (65 and over); less educated people; males in general, together with inhabitants of Ionian Calabria
and Adriatic Apulia are placed below the average value.

---

[3] Multiple responses question, percentages are based on cases. The overall total can exceed 100%.





These data could be more usefully scrutinized by considering the main sources of knowledge which have been used by the people. Social images of the tsunami and in turn, risk characterizations rely on a variety of sources, combined in different patterns according to age, education, gender and coastal areas. We assumed that broadcast media, printed media, the Internet and other sources, including word of mouth through interpersonal networks (relatives, neighbours and friends) would have different impacts on people's understanding of tsunamis, thus resulting in different mental models. We asked people to list the sources of information they actually used from a list of 12 items (multiple response questions), thus analysing both the relative relevance of any single source and of their possible combinations. It turned out that television has a paramount relevance as first source of information for almost all the respondents, in line with general statistics on cultural consumptions in Italy (Istat, 2018).

With regards to the penetration rates of any single source, 87% of the interviewees gathered information from TV News, while 21.2% saw scientific programmes or documentaries on free and thematic channels (such as SuperQuark, Focus, NatGeo and so on), also highlighting a huge gap with regard to other media sources. If we consider the possible combinations of both TV news and documentaries, penetration rate rises to 89,4%, that is to say that any effective risk communications campaign must face with the overwhelming role of television. Newspapers are however used only by 35.2% of respondents and books by 21.3%. The Internet, on the other hand, is surprisingly placed at the fourth place, reaching only 17.5% of the overall sample. This result could be influenced by the low offer of contents in scientific and governmental web sites, and spur us to increase and improve such offer, Understanding how to do this is one of the goals of this study.

Despite the availability of information from the media, interpersonal networks (relatives, neighbours and friends) are still nowadays important sources of knowledge on tsunami for 13.4% of the interviewees. In the following positions, data showed a 9.9% for broadcast radio and 6.4% for movies. The role of scientific and institutional communication from the Civil Protection, scientific institutions and local authorities is much more limited: the accumulated percentage of all the listed sources together weighs only 8% of cases.

Grouping channels into homogeneous categories, the overwhelming role of television emerges even more clearly, as it is able to reach almost 9/10 of the sample (89.4%), followed at a distance by traditional broadcast media (newspapers, books, movies, radio), which weighs for just over half (50.8%) and then by the Internet (17.5%). Interpersonal networks, which include all the interpersonal channel such as friends, parents and other relatives, together with neighbours and personal acquaintances were found to be a relevant source for 13,4%, while Institutional and scientific sources together with other sources are placed at the lower steps of this ranking. It is therefore important for research Institutions and Civil Protection agencies to work in this field, trying to reach more people and giving the correct information about this risk.





**Fig. 2 Sources of knowledge on tsunami: channels and penetration rates**

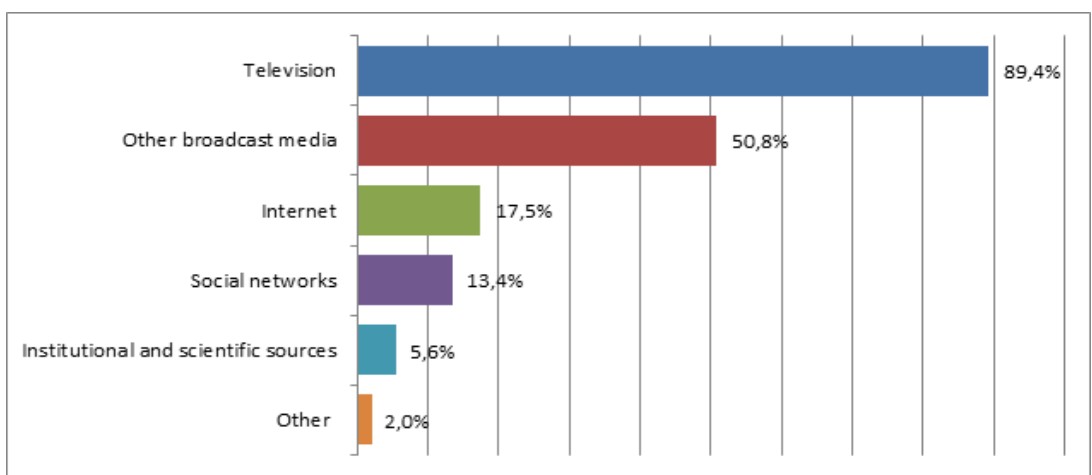

At a further level of analysis, one should also consider the number of sources that enter in individual's information
consumption and the patterns in which they are combined together, as to better understand how different social categories are
likely to draw onto specific pattern of usage, where sources are arranged and combined in different ways. The average number
of sources used is higher among graduates and postgraduates (2,67) and lower between those with a primary school certificate
or no certificate at all (1,48). Smaller differences, although relevant, were found for coastal slopes, so that people of
Tyrrhenian Calabria shows higher values with respect to other coastlines (2,55), while age has only limited influence on it
(ranging from 2.34 of the middle-agers to the 2.02 of over 65ers), and gender show almost equal values (2,22 men vs. 2.19
women).
Going deep into analysis, the way different channels are handled together can provide other relevant insight about the
relationship between interviewees and sources of knowledge: more than one third of the sample (34,6%) solely depends on
television, whereas the combination of television together with traditional broadcast media has been used by 29,8% and an
enriched combination of these two channels plus the Internet weighs for 8,1%. It is worth noticing how the accumulated
percentage of these first three combinations accounts for 72,5% of the whole sample, thus revealing a solid point of reference
to plan effective risk communication campaigns.
**Tab. 3: media sources used to gather information on tsunamis**

| | N | % Answers |
|---|---|---|
| Only television | 348 | 34,6 |
| Television + other broadcast media | 300 | 29,8 |
| Television + other broadcast media + Internet | 82 | 8,1 |
| Television + Internet | 43 | 4,3 |
| Only other broadcast media | 35 | 3,5 |
| Only interpersonal networks | 34 | 3,4 |
| Television + interpersonal networks | 34 | 3,4 |
| Television + other broadcast media + interpersonal networks | 22 | 2,2 |
| Other patterns (>2%) | 109 | 10,8 |
| Total | 100,0% | 100,0% |



As suggested by our results, the huge media coverage of the events of 2004 and 2011 in the Indian Ocean and Japan left a deep mark in social imagery of tsunami. Differently from the past, some of the most known pictures of such events came from digital eyewitness accounts, relayed through multiple internet channels, so that amateur user - generated contents quickly became the most important source of broadcast news from the most severely affected areas (Allan & Peters, 2015). Images such as those of big waves approaching the beach of Khao Lak (Thailand) after sea withdrawal, as well as well as the inundation wave exceeding seawalls and crashing on the seafront of Miyako, in Iwate prefecture (North-eastern Japan) went around the world, providing a vivid account of the event, contributing to shape people's understanding and mental models of tsunami at a global level. The absolute importance of such images has been shown in some papers (Yamori, 2013; Couling, 2014; Goeldner-Gianella et al. 2017), thus enhancing the impact of large magnitude, recent events with respect to the past ones (Wachinger et al., 2013).

Media role is not yet circumscribed to warning dissemination, as they are pivotal in enhancing risk perception, also playing an increasing role as knowledge mediators, raising risk awareness and improving preparedness (Romo-Murphy & Vos, 2014). Other research demonstrated that simply eliciting the word 'tsunami' was sufficient to enhance risk perception, also affecting motivational states and long term-decisions about health and benefit ratings (Västfjäll, Peters & Slovic, 2014). Media strategies to cope with disaster and effectively spread / disseminate tsunami alert messages are also incorporated in both research groups and Disaster Management Agencies' recommendations and documents across the world (Spahn et al., 2010; UNESCO, 2012).

**5.2 Area 2 (data): Tsunami risk perception**

Considering whole sample, the majority of people (49,5%) consider tsunami to be rather unlikely in the Mediterranean area. The occurrence of such an event is deemed to be unlikely (41%) or not at all likely (8,5%). By contrast, the overall percentage of those who think that tsunamis are not an odd and unrealistic event is 42,5%, and more exactly one third consider it quite likely (33%) and 9,5% holds it to be very likely, while 7,9% have no idea about its probability (see Fig. 3).

**Fig. 3 Perception of Tsunami occurrence in the Mediterranean area**

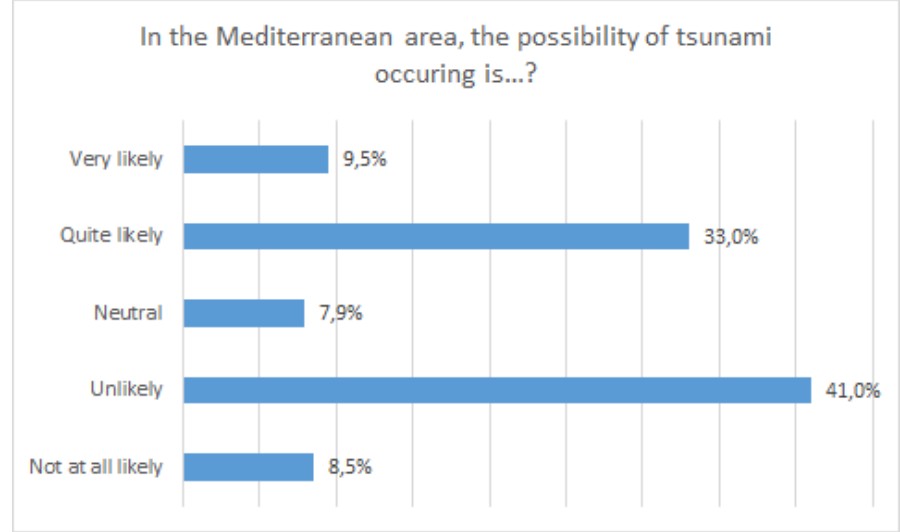




**Fig. 4 Which Mediterranean areas are perceived at a higher tsunami risk**

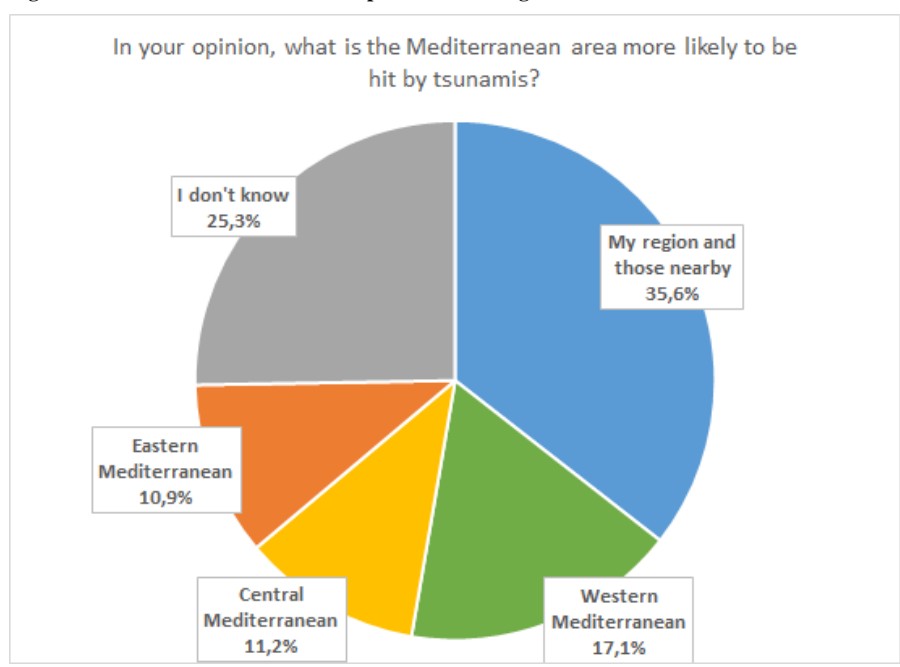


Fig. 4 show in detail which Mediterranean areas are deemed to be subjected to tsunami hazard. 35.6% of the sample indicates
their region of residence and nearby ones (Calabria and Puglia coasts), 17.1% the Western Mediterranean, the 10.9% the
Eastern Mediterranean, 11.2% the central Mediterranean and 25.3% simply don't know.
With respect to the geographical area of reference of the sample (Italy: Calabria and Puglia) the question: 'Do you think that
the coasts of your municipality could be hit by a tsunami?' has literally splitted the sample in half: those who answered Yes
were 44.7%, whilst No had 44.8%, and the remaining (10.6%) were not able to answer. The answers are slightly different
depending on the age group. Tsunami risk perception is slightly higher between respondents aged 50-64 (Fig. 5).



**Fig. 5 - Tsunami risk perception compared to the coasts of the municipalities of the respondents per gender and age groups.**

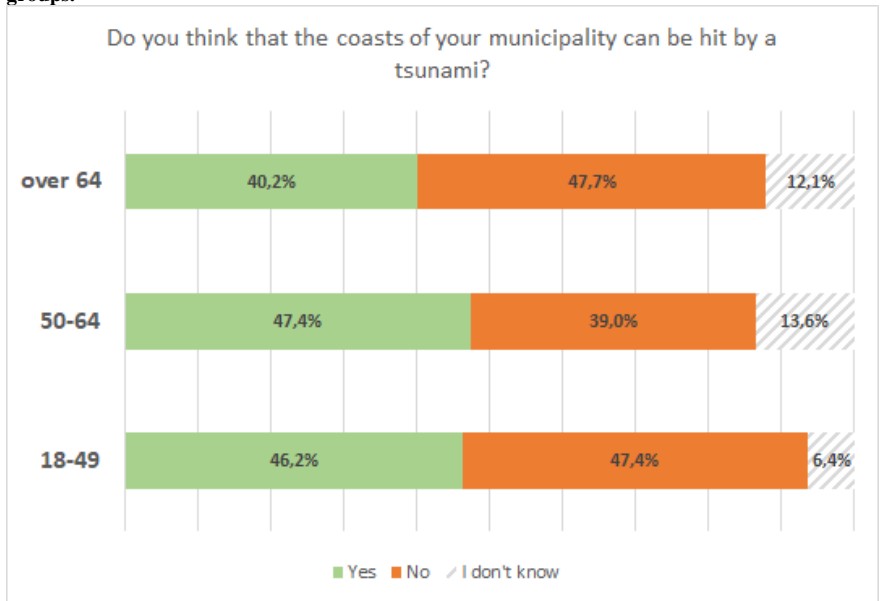

A similar situation arises when we compare the tsunami risk perception with the respondents' level of education. Risk perception grows along with education level, and graduate and post-graduates shown the higher percentage of those who consider the coasts of their municipality prone to tsunami risk (48%). (Fig. 5.2.4)

**Fig. 6 - Tsunami risk perception compared by education (L = Low level of education or no instruction; I = Intermediate, secondary school and high school; H = graduate and post-graduate).**

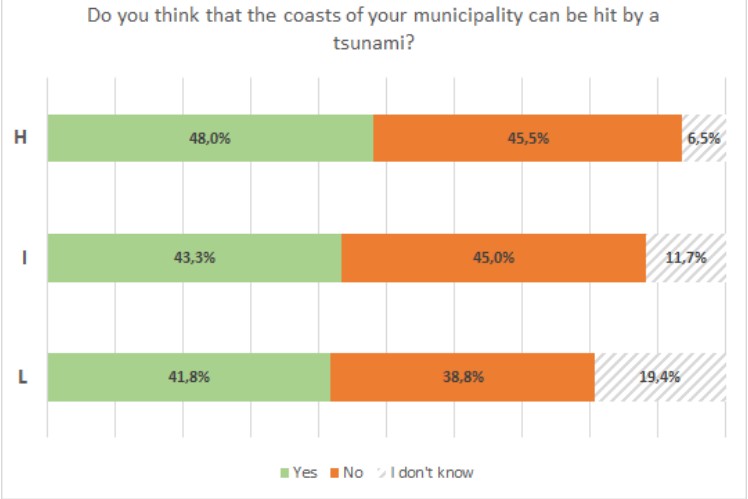

Fig. 7 shows results of tsunami perception for number of generations of residence in the considered coastal municipality. It can be noted that, contrary to what happens for the age, the number of generations of residence has a small effect on tsunami risk perception. This may indicate that considering the whole sample there is a low transfer of information and experiences




related to the tsunami risk from a generation to the following ones, while local events occurred in a more recent past may
trigger other patterns of information gathering and seeking.
Collective memory of natural disaster is a relevant issue to keep in consideration: although a single definition is still missing,
some authors highlight it as a dynamic process of functional adaption to a changing world (Assmann, 1997). Communication
plays a key role, because it allows memories to circulate, also connecting historically separate generations that otherwise
could not have mnemonic access to each other. This mnemonic transitivity allows people to preserve memories in the form of
oral traditions, passed on from one generation to the next one by mean of elders and families (Nora 1984-1986). The
transmission of such forms of collective memory is sometimes fostered through an institutionalisation process, that is the way
risk mitigation measures are incorporated into stable, accepted practices which are subjected to a specific regulation within a
given legal framework. Such institutionalisation processes are complementary to practices handed down through oral cultures:
awareness and resilience are enhanced by pushing social institutions (through school, public events, media campaigns) to
cultivate memory of past events using traditional stories and specific events. Two relevant examples of this come from
Indonesia and Japan. A traditional Indonesian song on tsunami and related mitigation measures (the smong song) was issued
in 1907 after a large, catastrophic event, allowing people living in Simeulue, Indonesia, to miraculously escape death from
2004 Boxing Day Tsunami of Banda Aceh (McAdoo et al., 2006). In a similar way, Japanese schoolchildren learn about
tsunami risk through educational stories about the wise Goryo Hamaguchi's, who saved his community from an ongoing
tsunami after the 1854 Ansei – Nankai Earthquake (Nishikawa, & Hosokawa, 2015).

**Fig. 7 - Tsunami risk perception compared to the number of generations of residence in the area (% column)**

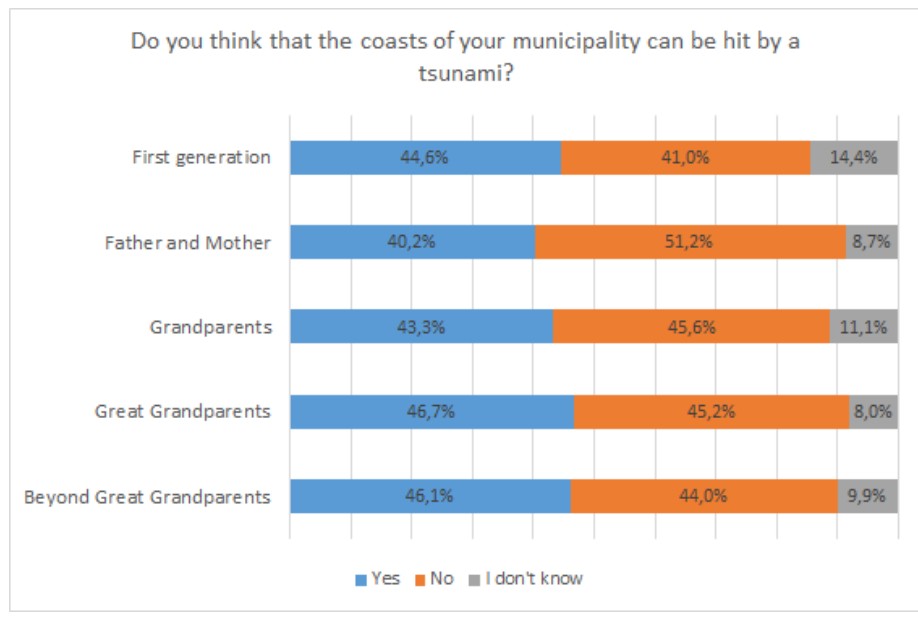


A peculiar aspect regards the perception of risk with respect to the different coastal regions. In this case, as shown in Fig. 8,
Tyrrhenian Calabria has a risk perception level higher than the others coasts.
At the moment, we are not able to explain this particular result with respect to the perception of tsunami risk in the Tyrrhenian
coast of Calabria, but we can formulate some hypotheses. We noted that respondents of such a coastal area are also more
likely to consider volcanoes as a possible tsunamigenic source: 66,2% of them indicates volcanoes as a possible cause of the



phenomena, while people from other coastal areas are far below this value, varying from a minimum of 41,4% for those living
in Adriatic Apulia to 45% of residents in Ionian Calabria. Indeed, the southern Tyrrhenian sea hosts several active and
quiescent volcanoes, including the Aeolian Islands (Stromboli, Vulcano), and submerged volcanoes like the Marsili, Palinuro
and other sea mounts (Figure 1). Therefore, the results outlined above could reflect both people's knowledge of this presence
(in bright days people living on the coasts of Tyrrhenian Calabria can see the volcanoes off shore), and the fear of submarine
eruptions and tsunamis, particularly from Mt. Marsili, to which a strong devastating power is attributed often by media.
Moreover, it must be considered that a 'volcanic' tsunami did actually occur in 2002 triggered by a collapse of the Sciara del
Fuoco flank, on Stromboli island, with run-up as high as 10 m in the island and notable effects even in Calabria (Bonaccorso
et al., 2003: Maramai et al., 2005; Tinti et al., 2005; Chiocci et al., 2008). These results need to be better assessed in the light
of multivariate analysis and possibly deepened through a specific qualitative research project on this area.
**Fig. 8 - Tsunami risk perception versus coastal regions.**

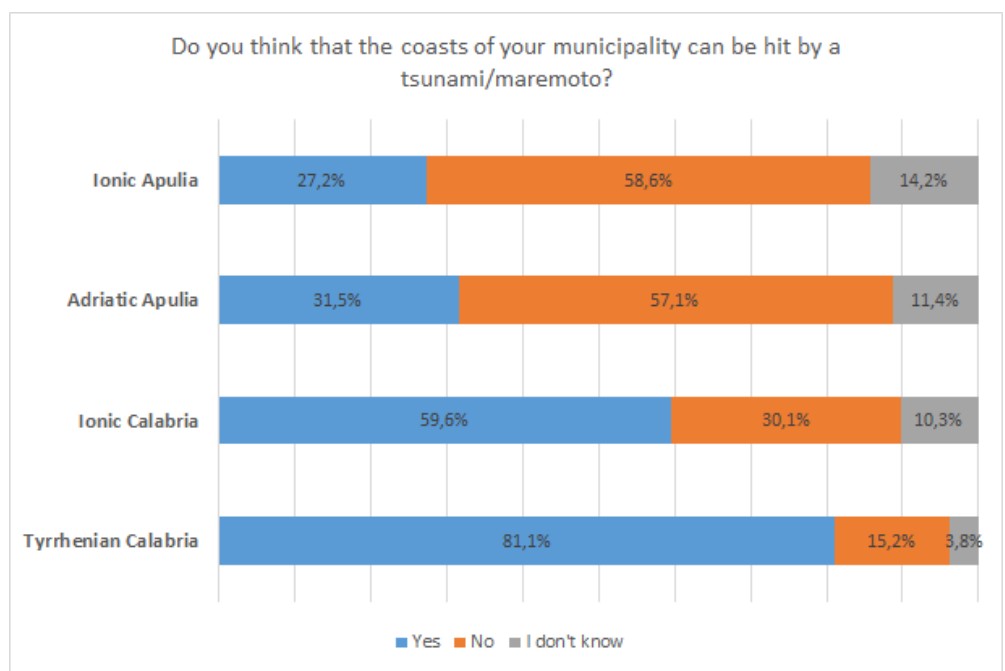

With regard to the perceived hazard of a tsunami waves, it should be borne in mind that tsunamis may come in all shapes and
colours, and even a small event can result in serious damages and loss of life (such as dragging into the sea both children and
adult persons). Despite the higher probability of occurrence of small tsunamis, and the huge hazard posed by waves of less
than a meter (with velocity up to 10m/s), people are still likely to refer to tsunamis being influenced by the strong and
persistent imagery of big events displayed on television. Only 16% of the sample consider that a wave size of 50-100 cm
would be hazardous for an adult staying near the shore, and those who think that also smaller tsunami waves could be a
serious threat are even less: just a 3,2%.
**Tab. 4: how high should the water level rise to be dangerous for people near the shore?**





|  | N | % |
| --- | --- | --- |
| Less than 50 centimetres | 33 | 3,2 |
| Between 50 centimetres and 1 meter | 163 | 16,0 |
| Between 1 and 3 meters | 357 | 35,0 |
| Over 3 meters | 402 | 39,4 |
| Don't know | 66 | 6,5 |
| Total | 1021 | 100,0 |


## 6. Conclusions

Empirical data showed that tsunami risk is generally underrated, deemed to be very unlikely for one individual to see in the
course of his own life. The level of risk perception seems to be quite low for the whole sample, and it appears being
influenced by education level and gender, as well as the possibility to access reliable sources of information.
An interesting result emerged from this study is that the inhabitants of the coastal area of Tyrrhenian Calabria are more likely
to consider tsunamis as actual and impending threats. As discussed in the previous section, this might be related to real or
purported volcanic risk from Aeolian Island or other possible eruptions from submerged volcanoes (Figure 1). Such a
circumstance would suggest the need of a thorough analysis on cultural and historical factors that may locally affect the way
tsunami risk is perceived and understood.
Mental models of tsunamis, stemming from people's characterization of hazard, appear to be heavily influenced by media
images of Sumatra (2004) and Japan (2011) devastating tsunamis, since TV news coverage and documentaries of these event
are the first source of information in terms of importance for most of our interviewees. Both disasters received a huge media
coverage, triggering a global-scale 'media event", where massive media audiences are brought out from daily routine and
concerns, being involved in highly ritualized pattern of media consumption until becoming a single, global community, held
together by the same mediated experience of the event (Dayan & Katz, 1992), which deeply shaped individual and social
understanding of tsunami.
Television is definitely at the core of 'modern' experience of 'distant' disasters, and tsunamis are no exception. The redundant
pictures of unbridled force, inconceivable destruction, death and suffering went around the world for many years, becoming a
visual paradigm of the tsunami itself. Evidence from our survey provides a robust support to this interpretive hypothesis: the
way tsunamis are understood is very consistent with such a televised imagery, and almost nine people out ten cite such media
channel as a primary source of information.
Risk characterization, which resumes the way hazard are understood, is affected by different factors, including the words that
are used to refer to such phenomena. Our results highlight that some features of this event are differently conceived when
using the exotic word 'tsunami' rather than the Italian word 'maremoto". Although the two terms are equivalent for Italian
Earth scientists, according to people's perception the two words refer to two different events, with some features in common.
Our results also show that people appear to be conscious that earthquakes are the most frequent cause of tsunamis. Also, they
tend to overestimate volcanoes as a possible cause of tsunamis, while underscoring other causes such as landslides. Anyway,
there is a poor awareness of some aspects of such a hazard: previous disasters in Italy are in any case part of a distant past,
whose details are doomed to fade away. Moreover, media accounts totally neglect possible impacts of small tsunamis, thus
fostering a false sense of subjective immunity.

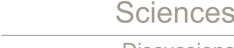
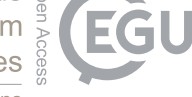
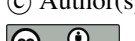

Research data made emerge a critical point: people are likely to match information on tsunamis with their personal experience
about sea-storm waves to understand and characterize such a risk, thus resulting into misleading assumptions about the hazard
posed by tsunamis. Recent studies have shown how people in different countries are likely to minimize the threat posed by
relatively small waves (up to the height of 3m) also underestimating the risk posed by bigger waves than those observed in
recent events. Similar phenomena have been also observed in countries recently hit by catastrophic tsunamis, posing serious
issues about people willingness to evacuate in case of an event (Oki & Nakayachi, 2012; Santos et al. 2016; Sutton & Woods,
2016; Wood et al., 2018).
**§ The value of a research**
This research aims at integrating and enriching tsunami-related literature from social sciences fields, also providing new data
and insights on the Mediterranean area. Currently, most of the available contributions regard only a few coastal areas in
Pacific and Indian Ocean, such as Japan, Indonesia, Chile, Cascadia and Pacific Islands, where tsunamis are considered both
as a matter of fact and a historical reality, where the risk posed by tsunamis is fairly known by local populations. To date,
research papers on this topic are noticeably scarce for the NEAM area, with a few local exception, for example some
Norwegian fjords (Lacasse and Nadim, 2011, Rød et al., 2012; Goeldner-Gianella et al., 2017). Lacking directions on people's
perception and understanding of what is a tsunami and its related damages may lead to significant difficulties in setting-up
sound risk communication strategies. Furthermore, the lack of data from social science could result in serious difficulties in
fostering people's engagement and participation in the implementation of effective mitigation measures. In general terms, the
development of tsunami warning systems should not focus only on managing an ongoing event through crisis communication,
but it should improve individuals' and communities' awareness and preparedness in the long term run (Lundgren and
McMakin, 2008). This implies a better understanding of targets, messages and channels to be arranged both for informing
people about the hazard posed by tsunamis, and to effectively shape an alerting strategy, where people are already conscious
about what it is happening and what they should do in case of an event.
**§ Innovativeness; originality; scientific rigour**
The research is first intended at providing viable knowledge about people perception and attitudes toward tsunami related
risks, to improve communication strategies of both CAT - INGV and Civil Protection Department (DPC) also providing
useful cues and suggestion to the overall Tsunami community in Mediterranean Region and beyond. This is the first extensive
study on tsunami risk perception in Italy, and the first of this kind (with large stratified sample and CATI interviews) being
completed in the NEAM region.
Any effective, sound risk communication strategy should lie on the integration of theory, empirical research, best practices
and careful assessment of outcomes, within an open ended cycle of research and action. Research results may indeed foster an
open discussion on risk and crisis communication strategies to be held, as to improve both individual awareness and
communities' involvement and participation to risk reduction programs at national and regional level.
**§ Implication for risk communication**
Risk communication should be integrated with other community engagement initiatives rather than being conceptualized as a
stand-alone process. The relevance and the meaning of the information about tsunamis arise from the way they are interpreted
and prioritized within given social contexts, hence any successful communication strategy must consider if and how
information is known and whether it is used, to facilitate preparedness (Paton et al. 2008). In particular, it would be important



to challenge commonplaces about tsunami, consider actual knowledge and education level of whose live in tsunami prone
areas, always bearing in mind the channel to be used to reach as many people as possible.

**§ Limitations and further developments**
The validity of the data collected and analysed in this paper is limited by definition to the coastal populations of Calabria and
Apulia, and cannot be generalized indiscriminately to the entire Italian coastal population or elsewhere. The general structure
of the questionnaire, the type and number of questions, as well as the duration of the interviews are strictly designed to be
administered via telephone.
Survey methodology entails an implicit assumption: data about individuals are used to make inferences about social attitudes
and beliefs, thus underestimating the influence of both local culture and 'group thinking' when facing complex problems. For
these reasons the survey should be ideally seen as a first step in a wider research strategy, aimed at providing further
developments within a mixed-method approach, 'to bring in more robust evidence than either qualitative or quantitative
approaches provide when they are used separately' [...] and 'to gain a deeper understanding of hazard perception and
preparedness' (Alam, 2016: 158).
The research is indeed conceived as a set of integrated modules, to fit different needs and social context, and it is suitable to
be replicated as a whole or in part in other geographical contexts, both in Italy and in the countries of the North East Atlantic,
Mediterranean and connected sea region (hereinafter NEAM). Data comparison and multivariate analysis may reveal
underlying cross-cutting factors of tsunami risk perception predictors, thereby focusing similarities and differences between
different coastal areas and countries.
Research with non-standard techniques (focus, interviews, collection of biographies) on specific target groups may also
complement research, as to clarify the role of both culture and individual motivations in shaping social response and risk
awareness.




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
