# Peer review of "Tsunami risk perception in Southern Italy: first evidence from a sample survey."

_Natural Hazards and Earth System Sciences, 2019_

## Referee Comment (RC1) · Anonymous Referee #1 · 8 May 2019

The research presents a gap in the literature regarding the risk perception of citizens who did not recently experience a tsunami or those who think that an event like this will never occur in the Mediterranean ocean. I believe this is an interesting topic and it has scientific significance. Nevertheless, there are major issues in the manuscript that the authors must address to be suitable for publication.

State three objectives is a big risk. According to the results of the article, the authors only address the first objective. The authors must delimit the scope of the paper. The three goals probably can be turned into three different papers.

The hypotheses are not clear. The first RH1 must be redefined as a hypothesis and not as an affirmation. The RH2 is not relevant. Many studies already discover these differences.

[Figure]

The methods and techniques section is deficient. A description of the study area and sample characteristics are missing. Also, there is not a description of the questionnaire, and most importantly, there is no evidence regarding the questions, neither the papers that were used to select the questions. The authors perform a focus group to test the questionnaire? Which was the no response rate?

There are many errors about the numbers of the figures and tables, and many of them were not used in the text, such as Figure1, Table 1 and Figure 6.

The Discussion section must be stated as "Results and Discussion" because it is confusing to the reader a Discussion section with so many results in it. Finally, I really think that the paper has significant relevance for the area, but the authors must rewrite the manuscript and organize it according to the journal standards.

---

## Referee Comment (RC2) · Anonymous Referee #2 · 24 May 2019

This paper is relevant and addresses an area where there is a gap of knowledge, in the Mediterranean and other places in the world, especially where tsunami are infrequent, but could also be of high impact. I feel this paper provides the state of awareness of tsunamis in the region under consideration that would be helpful for implementing disaster risk reduction strategies.

The questions used in the survey should be included as a supplement.

The section on Research Hypothesis with the two Research Hypothesis needs to be rewritten and stated more clearly. There is a reference to Mitigation measures, but the paper does not address the state of mitigation (preparedness) efforts in the region.

In the interpretation of the findings, there is no reference to preparedness and edu-

cation outreach activities that have been carried out and may lead to a different risk perception, in addition to the presence of the volcanoes.

Need to fix numbering of the Figures and verify reference to them in the text.

The map of with distribution of interviewees, needs to have clearly labeled the places referred to in the text. It would also be helpful to see on this or another map, areas that have been the source of have been impacted by previous events and are referred to in the text.

I was very confused by what was lumped together under "other broadcast media" in figure 2 and Table 3 (which the first column is not added up correctly) - it does not match the narrative. Did the question on INTERNET, also include Social Media specifically? For many Internet (web site) is very different from social media. I am suprised to not see a Social Media category.

In 5.2 and Figure 3 it is not clear to me the interpretation of neutral? In the text it says for this category respondent "had no idea about its probablity", is this really the case, it seems to be that the intermediate between Quite Likely and Unlikely would be "likely" or does neutral mean "i dont know". This is important because it affects the conclusion with regards tpo the state of perception.

The Conclusion section needs to be rewritten and be more substantial with a focus on the findings from the survey. The authors go off on tangents, that are not related or a product of the surveys.

One of the areas for further development would seem to be to integrate the perception of the tourists, which account for a greater number than residents.

It refers to focus groups as non standard, it was my impression (I am not a social scientist) that these were valid. Many of the social science studies our agency supports use focus groups. What is meant by "collection of biographies".

The English language is not of good quality in many sections, especially the Abstract,

Introduction and conclusion.

The second part of the title is not clear/compelling. It uses the term "sample suvey" - a survey is always a sample....

---

## Author Comment (AC1) · 22 Jul 2019

Authors' final response

Dear Editor,

we would like to thank you and the two reviewers for the very valuable suggestions. We attach below our point-to-point answers to Referees' comments. We do not send an updated version of the manuscript, but we are ready to revise the whole text whenever we will be asked to proceed.

Anonymous Referee 1

- Anonymous Referee 1 The research presents a gap in the literature regarding the

risk perception of citizens who did not recently experience a tsunami or those who think that an event like this will never occur in the Mediterranean ocean. I believe this is an interesting topic and it has scientific significance. Nevertheless, there are major issues in the manuscript that the authors must address to be suitable for publication.

State three objectives is a big risk. According to the results of the article, the authors only address the first objective. The authors must delimit the scope of the paper. The three goals probably can be turned into three different papers.

- Authors response We will duly take into account this suggestion. The first objective is comprehensive enough to set a discussion and objectives 2 and 3 will be briefly discussed as a corollary of objective 1.

- Authors' proposed changes (with line numbers) [Lines 106-111] This pilot study has the main strategic goal of providing empirical data on citizens' understanding and risk perception in a tsunami risk prone area, also allowing future comparisons with different areas of the NEAM Region. Moreover, the results will contribute to identify key messages, channels and techniques to effectively communicate tsunami risk in the Mediterranean area.

- Anonymous Referee 1 The hypotheses are not clear. The first RH1 must be redefined as a hypothesis and not as an affirmation. The RH2 is not relevant. Many studies already discover these differences.

- Authors response We will rewrite hypothesis 1 and 2. We better focused RH1. We accept referee's suggestion about RH2 to better focus the way tsunami risk is perceived in the different coastal areas, also considering the influence of the social demographic variables.

- Authors' proposed changes (with line numbers) [Lines 227-231] RH1: Does people's perception in Italy about tsunamis rely upon media representations of catastrophic events such as those occurred in Sumatra and Japan? RH 2: Are there differences

in risk perception related to different coastal areas and/or hazard level?

- Anonymous Referee 1 The methods and techniques section is deficient. A description of the study area and sample characteristics are missing. Also, there is not a description of the questionnaire, and most importantly, there is no evidence regarding the questions, neither the papers that were used to select the questions.

- Authors response We agree. We will provide a wider and stringent description of the areas and of the reference universe, of sampling methods and sample characteristics, including response rate. We will also provide an in-depth account of the way questionnaire was built up, including a discussion of the literature we have used. The main reference is the Dominey-Bird and Howes papers, which are cited in the reference list, but we also opted to personalise the questions. The way questionnaire has been developed and tested will be also described in this section.

- Anonymous Referee 1 The authors perform a focus group to test the questionnaire? Which was the no response rate?

- Authors response We will better describe the way questionnaire was built. We organised and administered two focus groups with 1) scientists and 2) lay people. The first one involved INGV tsunami scientists for a first review and an elicitation of scientific content of the questions. The aim of this focus group was to ensure that questions would have properly translated physical measures inherent to tsunamis (such as Maximum Inundation Eight, ingression and so on) into questions comprehensible for lay people. Our goal was to address in the most precise way the gap between "(mediated) representation of tsunamis" and "physical reality of tsunamis". Secondly, the questionnaire was tested on a non – representative sample of about twenty people with socio - demographic characteristics closer to the sample, to assess questions' readability, understanding and possible bias in the way questions were formulated. More in detail, we submitted the questionnaire to people with low and medium level of education rather than graduates, then we asked them to provide feedback. Suggestions were used as

a base to rephrase some questions as to make them easier to understand.

- Authors' proposed changes (with line numbers) [Line 235] we will add the previous two paragraphs in the paper.

- Anonymous Referee 1 There are many errors about the numbers of the figures and tables, and many of them were not used in the text, such as Figure1, Table 1 and Figure 6.

- Authors response We apologize for this. We will fix the problems.

- Anonymous Referee 1 The Discussion section must be stated as "Results and Discussion" because it is confusing to the reader a Discussion section with so many results in it.

- Authors response We agree. We will rename and re-arrange this section to make it more readable.

- Anonymous Referee 1 Finally, I really think that the paper has significant relevance for the area, but the authors must rewrite the manuscript and organize it according to the journal standards.

- Authors response We will thoroughly revise the manuscript according to the journal's guidelines.  

Please also note the supplement to this comment:
https://www.nat-hazards-earth-syst-sci-discuss.net/nhess-2019-97/nhess-2019-97-AC1-supplement.pdf

———————————————

[Figure]

**Supplement:**

[Figure]

**Survey on risk perception of tsunami of seismogenetic origin**

**Presentation Formula**

*Good Morning, good evening, on behalf of the National Institute of Geophysics and Volcanology, we are carrying out a research on a phenomenon called tsunami. If you agree to answer, I will ask you some questions on this issue. The information we are going to collect will allow both the National Institute of Geophysics and Volcanology and the other institutions of the Civil Protection System to better understand the this type of phenomena is perceived by the population and thus prepare the most appropriate interventions.*

*Please note that the information you provide will not be associated in any way with your name, but will be treated anonymously and statistically aggregated. The interview will last a maximum of 10 minutes. Thank you for your availability*

**SECTION 1: SOCIO-DEMOGRAPHIC DATA AND INFORMATION ON RESPONDENTS' BELONGING TERRITORY**

**Province** |__|__|

**Municipality** ________________________

1) **Gender**   ❑ M   ❑ F      ❑ Not willing to declare it

2) **Age**      |__|__|

3) **Education**
❑ Elementary licence
❑ Middle school licence
❑ Diploma of completion of secondary education
❑ University Degree
❑ Master's degree, PhD, other post-lauream titles
❑ No qualification

4) **Nationality**
❑ Italian
❑ Non-Italian

5) **How is your family household composed?** (multiple responses allowed)
- ❑ I live alone
- ❑ With adult(s)
- ❑ With children under 6 years
- ❑ With children under 6 years
- ❑ With elders
- ❑ With disabled / reduced mobility persons

6) **How many kilometres far from the shore do you live?**
- ❑ Within 1 km
- ❑ From 1 to 3 km
- ❑ Over 3 km
- ❑ I don't know

7) **How many generations is your family living on this coastline?** (Even in different households in the same area)
- ❑My family was here from before my great-grandparents
- ❑My great-grandparents lived there
- ❑My grandparents lived there
- ❑My father or my mother lived there
- ❑We are the first generation settled here

**SECTION 2: LEVEL OF AWARENESS AND SOURCES OF KNOWLEDGE ABOUT TSUNAMI HAZARD**

8) **Have you ever heard about  tsunami?**
- ❑ Yes ❑ No                                                    (if not, go to question 14.)

9) **Can you tell me which words do you associate with tsunami?**
❑Earthquake; ❑seaquake; ❑large wave; ❑port; ❑flooding; ❑rough sea; ❑inundation; ❑sea withdrawal; ❑currents; ❑strong wind.

10)    **Can you tell me which words do you associate with seaquake?**
❑Earthquake; ❑seaquake; ❑large wave; ❑port; ❑flooding; ❑rough sea; ❑inundation; ❑sea withdrawal; ❑currents; ❑strong wind.

11)    **Can you tell me which word sound more familiar to you?**
❑Tsunami  ❑Seaquake

12) **If you already heard about tsunami, from which sources?** (multiple responses allowed)

❑ Books
❑ Newspapers
❑ Tv News
❑ Radio
❑ Internet
❑ Television programs / channels about science (SuperQuark, Focus, NatGeo etc.)
❑ Movies
❑ Civil Protection (#IoNonRischio – Maremoti, drills).
❑ Region, provinces or municipalities
❑ Research institutions / Universities
❑ Friends / relatives
❑ Other (specify)
_______________________________ Other

13) **In the Mediterranean Sea the possibility of a tsunami / tidal wave is:**

| Not probable | Somewhat improbable | Neither improbable nor probable | Somewhat probable | Very probable |
|:---:|:---:|:---:|:---:|:---:|
| ❑ | ❑ | ❑ | ❑ | ❑ |

14) **In your opinion, what are the causes that may trigger a tsunami / seaquake?** (multiple responses allowed)

❑ Earthquakes
❑ Volcanic Eruptions
❑ Landslides
❑ Meteorites or asteroids at sea
❑ Weather phenomena
❑ Other (specify)
_________________________________________ Other
❑ I don't know

**SECTION 3: CONTEXTUAL PERCEPTION OF TSUNAMI HAZARD**

**15)** **The coast of your municipality are mainly**

❑ Low and sandy ❑ High and sandy
❑ Low and rocky ❑ High and rocky
❑ With inlets / coves ❑ Withou inlets / coves

**16)** **Do you think that the coasts of your municipality / province can be affected by a tsunami / seaquake?**

❑Yes ❑ No ❑I don't know

**17)** **In your opinion, which areas of the Mediterranean are more likely to be hit by tsunami / seaquake?** (multiple responses allowed)

| | |
|---|---|
| My own region or closer ones (Molise, Puglia, Calabria, Calabria, Sicily, Campania, Lower Tyrrhenian Sea, Gulf of Taranto) | ❑ |
| Western Mediterranean (Spain, Morocco, Algeria) | ❑ |
| Central Mediterranean (Italy, France, Slovenia, Croatia, Montenegro, Bosnia Herzegovina, Albania, Greece, Tunisia, Malta, Libya) | ❑ |
| Eastern Mediterranean (Coast Egypt, Israel, Lebanon, Syria, Turkey and Cyprus)? | ❑ |
| I don't know | ❑ |

**18)** **If a tsunami / seaquake would hit your region, how much do you think the water level could rise with respect to the normal sea level?**

❑ Less than 50 cm
❑ Between 50 cm and 1 meter
❑ Between 1 meter and 3 meters
❑ Over 3 meters
❑ I don't know

**19)** **If a tsunami / seaquake would hit your region, how far from the shore could the water arrive?**

❑ Less than 1 meter
❑ From 1 to 10 meters
❑ From 10 to 99 meters
❑ From hundred meters to kilometers
❑ I don't know

20) **In your opinion, in case of tsunami / tsunami, to be dangerous for people near the shore how much should the water level with respect to the normal sea level?**

- ❑ Less than 50 cm
- ❑ Between 50 cm and 1 meter
- ❑ Between 1 meter and 3 meters
- ❑ Over 3 meters
- ❑ I don't know

21) **What kind of phenomena may precede the arrival of a tsunami / seaquake?**
(How far do you agree with the following statements?)

❑ A strong or long-lasting earthquake

| Strongly disagree | Somewhat disagree | Neither agree nor disagree | Somewhat agree | Strongly Agree |
|---|---|---|---|---|
| ❑ | ❑ | ❑ | ❑ | ❑ |

❑ Sea withdrawal

| Strongly disagree | Somewhat disagree | Neither agree nor disagree | Somewhat agree | Strongly Agree |
|---|---|---|---|---|
| ❑ | ❑ | ❑ | ❑ | ❑ |

❑ Sea level rise

| Strongly disagree | Somewhat disagree | Neither agree nor disagree | Somewhat agree | Strongly Agree |
|---|---|---|---|---|
| ❑ | ❑ | ❑ | ❑ | ❑ |

❑ Strong sea currents

| Strongly disagree | Somewhat disagree | Neither agree nor disagree | Somewhat agree | Strongly Agree |
|---|---|---|---|---|
| ❑ | ❑ | ❑ | ❑ | ❑ |

❑ A strong and long-lasting uproar

| Strongly disagree | Somewhat disagree | Neither agree nor disagree | Somewhat agree | Strongly Agree |
|---|---|---|---|---|
| ❑ | ❑ | ❑ | ❑ | ❑ |

22) **Try to figure out the effects of a tsunami / tsunami on the coasts of your region. How far do you agree with the following statements?**

❑ Deaths or serious injuries

| Strongly disagree | Somewhat disagree | Neither agree nor disagree | Somewhat agree | Strongly Agree |
|---|---|---|---|---|
| ❑ | ❑ | ❑ | ❑ | ❑ |

❑

Damage / destruction of households, buildings or infrastructures

| Strongly disagree | Somewhat disagree | Neither agree nor disagree | Somewhat agree | Strongly Agree |
|---|---|---|---|---|
| ❏ | ❏ | ❏ | ❏ | ❏ |

❏Impacts on natural environment (changes in coastal landscape, impacts on vegetation, crops, wild animals and livestock, etc.)

| Strongly disagree | Somewhat disagree | Neither agree nor disagree | Somewhat agree | Strongly Agree |
|---|---|---|---|---|
| ❏ | ❏ | ❏ | ❏ | ❏ |

❏Impacts on the economy and work (destruction of coastal enterprises, damages to tourism industry, scarcity of financial resources, etc.)

| Strongly disagree | Somewhat disagree | Neither agree nor disagree | Somewhat agree | Strongly Agree |
|---|---|---|---|---|
| ❏ | ❏ | ❏ | ❏ | ❏ |

**SECTION 4: REPRESENTATION OF TSUNAMIS**

23) **How far do you agree with the following statements?**

❏ When you feel a strong earthquake near the coast one should flee to the beach

| Strongly disagree | Somewhat disagree | Neither agree nor disagree | Somewhat agree | Strongly Agree |
|---|---|---|---|---|
| ❏ | ❏ | ❏ | ❏ | ❏ |

❏ A tsunami waves can flood the coastal inlands for kilometers

| Strongly disagree | Somewhat disagree | Neither agree nor disagree | Somewhat agree | Strongly Agree |
|---|---|---|---|---|
| ❏ | ❏ | ❏ | ❏ | ❏ |

❏ A half-metre tsunami wave can drag an adult man into the sea

| Strongly disagree | Somewhat disagree | Neither agree nor disagree | Somewhat agree | Strongly Agree |
|---|---|---|---|---|
| ❏ | ❏ | ❏ | ❏ | ❏ |

❏ Sometimes tsunamis can be preceded by sea withdrawal, even for several tens of metres.

| Strongly disagree | Somewhat disagree | Neither agree nor disagree | Somewhat agree | Strongly Agree |
|---|---|---|---|---|
| ❏ | ❏ | ❏ | ❏ | ❏ |

❏ In the Mediterranean sea tsunami with waves higher than twenty meters are possible

| Strongly disagree | Somewhat disagree | Neither agree nor disagree | Somewhat agree | Strongly Agree |
|---|---|---|---|---|
| ❏ | ❏ | ❏ | ❏ | ❏ |

**SECTION 5: CULTURAL ATTITUDES AND WORLDVISION**

24) **How far do you agree with the following statements?**

❏ To move on in life you have to work hard and do what you are told to do

| Strongly disagree | Somewhat disagree | Neither agree nor disagree | Somewhat agree | Strongly Agree |
|---|---|---|---|---|
| ❏ | ❏ | ❏ | ❏ | ❏ |

❏ When a natural disaster occurs, authorities should impose severe and immediate punishment on those who break the rules

| Strongly disagree | Somewhat disagree | Neither agree nor disagree | Somewhat agree | Strongly Agree |
|---|---|---|---|---|
| ❏ | ❏ | ❏ | ❏ | ❏ |

❏ The healthiest society is the one based on competition between individuals

| Strongly disagree | Somewhat disagree | Neither agree nor disagree | Somewhat agree | Strongly Agree |
|---|---|---|---|---|
| ❏ | ❏ | ❏ | ❏ | ❏ |

❏ To deal with natural disasters individuals should prepare themselves before instead of waiting for someone to help them after

| Strongly disagree | Somewhat disagree | Neither agree nor disagree | Somewhat agree | Strongly Agree |
|---|---|---|---|---|
| ❏ | ❏ | ❏ | ❏ | ❏ |

❏ Many conflicts could be resolved by a fairer division of workload and natural resources

| Strongly disagree | Somewhat disagree | Neither agree nor disagree | Somewhat agree | Strongly Agree |
|---|---|---|---|---|
| ❏ | ❏ | ❏ | ❏ | ❏ |

❏ Natural disasters do not exist: it is only nature offended by man's arrogance and greediness, that rebels

| Strongly disagree | Somewhat disagree | Neither agree nor disagree | Somewhat agree | Strongly Agree |
|---|---|---|---|---|
| ❏ | ❏ | ❏ | ❏ | ❏ |

❏ When you do something for others you should not expect gratefulness

| Strongly disagree | Somewhat disagree | Neither agree nor disagree | Somewhat agree | Strongly Agree |
|---|---|---|---|---|
| ❏ | ❏ | ❏ | ❏ | ❏ |

DRAFT

❑ Natural disasters serve to remind us that making plans is useless: our lives are determined by forces that we cannot control

| Strongly disagree | Somewhat disagree | Neither agree nor disagree | Somewhat agree | Strongly Agree |
|---|---|---|---|---|
| ❑ | ❑ | ❑ | ❑ | ❑ |

**SECTION 6: MESSAGES AND CHANNELS TO SPREAD TSUNAMI ALERT**

25) **In your opinion, is it possible to spread an early warning before the arrival of a tsunami on the way to the coast?**
❑ Yes     ❑ No     ❑ I don't know (If answer is "not" or "I don't know" go to 25)

26) **In your opinion, who should warn population about the impending arrival of a tsunami on the way to the coast?**
1. _______________________________________________________________
2. _______________________________________________________________
3. _______________________________________________________________

27) **If a tsunami would hit your region's coastline, which channels would you prefer to receive an early warning?** (max three answers)

| |
|---|
| ❑ Television |
| ❑ Radio |
| ❑ Internet and social media |
| ❑ E-mail |
| ❑ Phone call (mobile or fixed telephone) |
| ❑ Be advised by friends, neighbours or relatives |
| ❑ SMS |
| ❑ Smarthpone application from official sources |
| ❑ Sirens or other acoustic devices |

---

## Author Comment (AC2) · 22 Jul 2019

- Anonymous Referee 2 This paper is relevant and addresses an area where there is a gap of knowledge, in the Mediterranean and other places in the world, especially where tsunami are infrequent, but could also be of high impact. I feel this paper provides the state of awareness of tsunamis in the region under consideration that would be helpful for implementing disaster risk reduction strategies.

The questions used in the survey should be included as a supplement.

- Authors response We agree and will provide both the Italian and English versions of the questionnaire as a supplement to the paper

[Figure]

- Anonymous Referee 2 The section on Research Hypothesis with the two Research Hypothesis needs to be rewritten and stated more clearly.

- Authors response We will rewrite hypothesis 1 and 2. We better focused RH1. We accept referee's suggestion about RH2 to better focus the way tsunami risk is perceived in the different coastal areas, also considering the influence of the social demographic variables.

Authors' proposed changes (with line numbers) [Lines 227-231] RH1: Does people's perception in Italy about tsunamis rely upon media representations of catastrophic events such as those occurred in Sumatra and Japan? RH 2: Are there differences in risk perception related to different coastal areas and/or hazard level?

- Anonymous Referee 2 There is a reference to Mitigation measures, but the paper does not address the state of mitigation (preparedness) efforts in the region.

- Authors response We hold that at present the efforts to improve awareness and preparedness in the region are ongoing, but they are neither sufficient nor satisfactory, despite National Civil Protection organized a number of campaigns and drills. From a legal and institutional perspective, the Italian tsunami early warning system was formally established only in 2017, and although it is almost fully operating in tsunami assessment and spreading early warning messages, a comprehensive risk communication strategy has not been implemented yet within the Italian civil protection system down to the local (municipality) level and to people (last mile issue is critical for most Tsunami Early Warning Systems [TEWS], especially the youngest ones). Sample survey was initially designed to provide INGV and Civil Protection with scientific evidence to better address a sound risk communication strategy. We will provide a general description of the activity named "Io Non Rischio" (I do not take risks) coordinated by Italian Civil Protection. If necessary we may provide a brief general description of such a campaign, also providing a link to the related website.

- Anonymous Referee 2 In the interpretation of the findings, there is no reference to

preparedness and education outreach activities that have been carried out and may lead to a different risk perception, in addition to the presence of the volcanoes.

- Authors response Research provides evidence of the minimal impact of institutional and scientific sources on people understanding of tsunamis, since previous outreach initiatives revealed to be neither numerous nor effective. Unfortunately, the number of interviewees who recalled the campaigns is too low to draw any significant statistical inference (N= 34/1021, i.e., about 3,3% of the sample). Moreover, "Io non Rischio" is a campaign conducted at municipality level and we have not enough answers to draw any conclusion at this level.

- Anonymous Referee 2 Need to fix numbering of the Figures and verify reference to them in the text.

- Authors response We agree and will fix the problem.

- Anonymous Referee 2 The map of with distribution of interviewees, needs to have clearly labeled the places referred to in the text. It would also be helpful to see on this or another map, areas that have been the source of have been impacted by previous events and are referred to in the text.

- Authors response We agree and will fix the problem.

- Anonymous Referee 2 I was very confused by what was lumped together under "other broadcast media" in figure 2 and Table 3 (which the first column is not added up correctly) - it does not match the narrative. Did the question on INTERNET, also include Social Media specifically?

- Authors response The notion of "other broadcast media" has been introduced to explain the consequences of different combinations of sources on risk awareness as these have been actually used by interviewees. The ways different sources are arranged into individuals' information gathering strategies provide different information with respect to the analysis of penetration rate for any single source. We will improve

the definition of any single combination of sources. As far as it emerged from the background research carried out prior the survey, the number and the quality of available sources on tsunami risk on Italian coastlines were found to be unsatisfactory. A clear-cut distinction between different sources within the internet appears to be difficult, since same contents are frequently disseminated through different platforms, such as blogs, web sites, social media, etc. Furthermore, the percentages for the whole category "Internet" were lower than one would have expected, thus making difficult to provide robust evidence from a more detailed categorization, in which any category is doomed to result in a smaller number of cases for each item (see the motivation about the campaigns). We will therefore aggregate residual categories (less than 4%) and will also provide a clearer description of the categories as of both the most relevant results and implications for risk communication strategies.

- Anonymous Referee 2 For many Internet (web site) is very different from social media. I am surprised to not see a Social Media category.

- Authors response Same as above: the whole number of cases is not sufficient to make further distinctions.

- Anonymous Referee 2 In 5.2 and Figure 3 it is not clear to me the interpretation of neutral? In the text it says for this category respondent "had no idea about its probability", is this really the case, it seems to be that the intermediate between Quite Likely and Unlikely would be "likely" or does neutral mean "I dont know".

This is important because it affects the conclusion with regards to the state of perception.

- Authors response The suggestion about using "neither likely nor unlikely" rather than "neutral" is right. We will change it in the figure. With regard to the difference between "Quite Likely" and "Unlikely" it refers to Renzis Likert scales theory and applications, and it is to be considered as a generally accepted standard in social research (see Likert, 1974. A method of constructing an attitude scale. in Scaling: A sourcebook for

behavioral scientists, 233-243.)

- Anonymous Referee 2 The Conclusion section needs to be rewritten and be more substantial with a focus on the findings from the survey. The authors go off on tangents, that are not related or a product of the surveys.

- Authors response We have modified the Conclusions eliminating some paragraphs, as suggested. We also added a final paragraph with some ideas for future research (see below).

Authors' proposed changes (with line numbers) [506—] This research is the first of this kind conducted in Italy. Its findings appear to be promising. Future analyses on this data set will probably allow us to better identify the main factors affecting tsunami risk perception in Italy, as well as to better understand the differences between different coastal areas. Future steps of this research include the extension to other contiguous coastal regions (namely, Basilicata, Molise, Sicily) which are most exposed to tsunami hazard, together with Calabria and Apulia.

- Anonymous Referee 2 One of the areas for further development would seem to be to integrate the perception of the tourists, which account for a greater number than residents.

- Authors response We fully agree on the need of specific research on this particular subject. Nevertheless, random sampling strategies are practically unfeasible with tourists. We also recognize the need and the opportunity to carry out a specific study to be administered with other techniques (e.g. focus group / random interviews).

- Anonymous Referee 2 It refers to focus groups as non standard, it was my impression (I am not a social scientist) that these were valid. Many of the social science studies our agency supports use focus groups. What is meant by "collection of biographies".

- Authors response We accept the solicitation: in Italian social science field the notion of standard / non standard methods and techniques refers to the difference between

statistical intensive techniques and interpretive / qualitative methods and it has been first introduced by the Italian methodologist Alberto Marradi, who was also recalled in some papers published by NHESS. We will better explain this distinction, holding that although focus groups are widely accepted, they actually do not provide statistical evidence.

- Anonymous Referee 2 The English language is not of good quality in many sections, especially the Abstract.

- Authors response We will hire a professional proof-reader to check the paper before submission.

Please also note the supplement to this comment:
https://www.nat-hazards-earth-syst-sci-discuss.net/nhess-2019-97/nhess-2019-97-AC2-supplement.pdf

**Supplement:**

[Figure]

**Survey on risk perception of tsunami of seismogenetic origin**

**Presentation Formula**

*Good Morning, good evening, on behalf of the National Institute of Geophysics and Volcanology, we are carrying out a research on a phenomenon called tsunami. If you agree to answer, I will ask you some questions on this issue. The information we are going to collect will allow both the National Institute of Geophysics and Volcanology and the other institutions of the Civil Protection System to better understand the this type of phenomena is perceived by the population and thus prepare the most appropriate interventions.*

*Please note that the information you provide will not be associated in any way with your name, but will be treated anonymously and statistically aggregated. The interview will last a maximum of 10 minutes. Thank you for your availability*

**SECTION 1: SOCIO-DEMOGRAPHIC DATA AND INFORMATION ON RESPONDENTS' BELONGING TERRITORY**

**Province** |__|__|

**Municipality** ________________________

1) **Gender**   ❑ M   ❑ F      ❑ Not willing to declare it

2) **Age**      |__|__|

3) **Education**
❑ Elementary licence
❑ Middle school licence
❑ Diploma of completion of secondary education
❑ University Degree
❑ Master's degree, PhD, other post-lauream titles
❑ No qualification

4) **Nationality**
❑ Italian
❑ Non-Italian

5) **How is your family household composed?** (multiple responses allowed)
- ❑ I live alone
- ❑ With adult(s)
- ❑ With children under 6 years
- ❑ With children under 6 years
- ❑ With elders
- ❑ With disabled / reduced mobility persons

6) **How many kilometres far from the shore do you live?**
- ❑ Within 1 km
- ❑ From 1 to 3 km
- ❑ Over 3 km
- ❑ I don't know

7) **How many generations is your family living on this coastline?** (Even in different households in the same area)
- ❑My family was here from before my great-grandparents
- ❑My great-grandparents lived there
- ❑My grandparents lived there
- ❑My father or my mother lived there
- ❑We are the first generation settled here

**SECTION 2: LEVEL OF AWARENESS AND SOURCES OF KNOWLEDGE ABOUT TSUNAMI HAZARD**

8) **Have you ever heard about  tsunami?**
- ❑ Yes ❑ No                                                    (if not, go to question 14.)

9) **Can you tell me which words do you associate with tsunami?**
❑Earthquake; ❑seaquake; ❑large wave; ❑port; ❑flooding; ❑rough sea; ❑inundation; ❑sea withdrawal; ❑currents; ❑strong wind.

10)    **Can you tell me which words do you associate with seaquake?**
❑Earthquake; ❑seaquake; ❑large wave; ❑port; ❑flooding; ❑rough sea; ❑inundation; ❑sea withdrawal; ❑currents; ❑strong wind.

11)    **Can you tell me which word sound more familiar to you?**
❑Tsunami  ❑Seaquake

12) **If you already heard about tsunami, from which sources?** (multiple responses allowed)

❑ Books
❑ Newspapers
❑ Tv News
❑ Radio
❑ Internet
❑ Television programs / channels about science (SuperQuark, Focus, NatGeo etc.)
❑ Movies
❑ Civil Protection (#IoNonRischio – Maremoti, drills).
❑ Region, provinces or municipalities
❑ Research institutions / Universities
❑ Friends / relatives
❑ Other (specify)
_______________________________ Other

13) **In the Mediterranean Sea the possibility of a tsunami / tidal wave is:**

| Not probable | Somewhat improbable | Neither improbable nor probable | Somewhat probable | Very probable |
|:---:|:---:|:---:|:---:|:---:|
| ❑ | ❑ | ❑ | ❑ | ❑ |

14) **In your opinion, what are the causes that may trigger a tsunami / seaquake?** (multiple responses allowed)

❑ Earthquakes
❑ Volcanic Eruptions
❑ Landslides
❑ Meteorites or asteroids at sea
❑ Weather phenomena
❑ Other (specify)
_________________________________________ Other
❑ I don't know

**SECTION 3: CONTEXTUAL PERCEPTION OF TSUNAMI HAZARD**

**15)** **The coast of your municipality are mainly**

❑ Low and sandy ❑ High and sandy
❑ Low and rocky ❑ High and rocky
❑ With inlets / coves ❑ Withou inlets / coves

**16)** **Do you think that the coasts of your municipality / province can be affected by a tsunami / seaquake?**

❑Yes ❑ No ❑I don't know

**17)** **In your opinion, which areas of the Mediterranean are more likely to be hit by tsunami / seaquake?** (multiple responses allowed)

| | |
|---|---|
| My own region or closer ones (Molise, Puglia, Calabria, Calabria, Sicily, Campania, Lower Tyrrhenian Sea, Gulf of Taranto) | ❑ |
| Western Mediterranean (Spain, Morocco, Algeria) | ❑ |
| Central Mediterranean (Italy, France, Slovenia, Croatia, Montenegro, Bosnia Herzegovina, Albania, Greece, Tunisia, Malta, Libya) | ❑ |
| Eastern Mediterranean (Coast Egypt, Israel, Lebanon, Syria, Turkey and Cyprus)? | ❑ |
| I don't know | ❑ |

**18)** **If a tsunami / seaquake would hit your region, how much do you think the water level could rise with respect to the normal sea level?**

❑ Less than 50 cm
❑ Between 50 cm and 1 meter
❑ Between 1 meter and 3 meters
❑ Over 3 meters
❑ I don't know

**19)** **If a tsunami / seaquake would hit your region, how far from the shore could the water arrive?**

❑ Less than 1 meter
❑ From 1 to 10 meters
❑ From 10 to 99 meters
❑ From hundred meters to kilometers
❑ I don't know

20) **In your opinion, in case of tsunami / tsunami, to be dangerous for people near the shore how much should the water level with respect to the normal sea level?**

- ❑ Less than 50 cm
- ❑ Between 50 cm and 1 meter
- ❑ Between 1 meter and 3 meters
- ❑ Over 3 meters
- ❑ I don't know

21) **What kind of phenomena may precede the arrival of a tsunami / seaquake?**
(How far do you agree with the following statements?)

❑ A strong or long-lasting earthquake

| Strongly disagree | Somewhat disagree | Neither agree nor disagree | Somewhat agree | Strongly Agree |
|---|---|---|---|---|
| ❑ | ❑ | ❑ | ❑ | ❑ |

❑ Sea withdrawal

| Strongly disagree | Somewhat disagree | Neither agree nor disagree | Somewhat agree | Strongly Agree |
|---|---|---|---|---|
| ❑ | ❑ | ❑ | ❑ | ❑ |

❑ Sea level rise

| Strongly disagree | Somewhat disagree | Neither agree nor disagree | Somewhat agree | Strongly Agree |
|---|---|---|---|---|
| ❑ | ❑ | ❑ | ❑ | ❑ |

❑ Strong sea currents

| Strongly disagree | Somewhat disagree | Neither agree nor disagree | Somewhat agree | Strongly Agree |
|---|---|---|---|---|
| ❑ | ❑ | ❑ | ❑ | ❑ |

❑ A strong and long-lasting uproar

| Strongly disagree | Somewhat disagree | Neither agree nor disagree | Somewhat agree | Strongly Agree |
|---|---|---|---|---|
| ❑ | ❑ | ❑ | ❑ | ❑ |

22) **Try to figure out the effects of a tsunami / tsunami on the coasts of your region. How far do you agree with the following statements?**

❑ Deaths or serious injuries

| Strongly disagree | Somewhat disagree | Neither agree nor disagree | Somewhat agree | Strongly Agree |
|---|---|---|---|---|
| ❑ | ❑ | ❑ | ❑ | ❑ |

❑

Damage / destruction of households, buildings or infrastructures

| Strongly disagree | Somewhat disagree | Neither agree nor disagree | Somewhat agree | Strongly Agree |
|---|---|---|---|---|
| ❏ | ❏ | ❏ | ❏ | ❏ |

❏Impacts on natural environment (changes in coastal landscape, impacts on vegetation, crops, wild animals and livestock, etc.)

| Strongly disagree | Somewhat disagree | Neither agree nor disagree | Somewhat agree | Strongly Agree |
|---|---|---|---|---|
| ❏ | ❏ | ❏ | ❏ | ❏ |

❏Impacts on the economy and work (destruction of coastal enterprises, damages to tourism industry, scarcity of financial resources, etc.)

| Strongly disagree | Somewhat disagree | Neither agree nor disagree | Somewhat agree | Strongly Agree |
|---|---|---|---|---|
| ❏ | ❏ | ❏ | ❏ | ❏ |

**SECTION 4: REPRESENTATION OF TSUNAMIS**

23) **How far do you agree with the following statements?**

❏ When you feel a strong earthquake near the coast one should flee to the beach

| Strongly disagree | Somewhat disagree | Neither agree nor disagree | Somewhat agree | Strongly Agree |
|---|---|---|---|---|
| ❏ | ❏ | ❏ | ❏ | ❏ |

❏ A tsunami waves can flood the coastal inlands for kilometers

| Strongly disagree | Somewhat disagree | Neither agree nor disagree | Somewhat agree | Strongly Agree |
|---|---|---|---|---|
| ❏ | ❏ | ❏ | ❏ | ❏ |

❏ A half-metre tsunami wave can drag an adult man into the sea

| Strongly disagree | Somewhat disagree | Neither agree nor disagree | Somewhat agree | Strongly Agree |
|---|---|---|---|---|
| ❏ | ❏ | ❏ | ❏ | ❏ |

❏ Sometimes tsunamis can be preceded by sea withdrawal, even for several tens of metres.

| Strongly disagree | Somewhat disagree | Neither agree nor disagree | Somewhat agree | Strongly Agree |
|---|---|---|---|---|
| ❏ | ❏ | ❏ | ❏ | ❏ |

❏ In the Mediterranean sea tsunami with waves higher than twenty meters are possible

| Strongly disagree | Somewhat disagree | Neither agree nor disagree | Somewhat agree | Strongly Agree |
|---|---|---|---|---|
| ❏ | ❏ | ❏ | ❏ | ❏ |

**SECTION 5: CULTURAL ATTITUDES AND WORLDVISION**

24) **How far do you agree with the following statements?**

❏ To move on in life you have to work hard and do what you are told to do

| Strongly disagree | Somewhat disagree | Neither agree nor disagree | Somewhat agree | Strongly Agree |
|---|---|---|---|---|
| ❏ | ❏ | ❏ | ❏ | ❏ |

❏ When a natural disaster occurs, authorities should impose severe and immediate punishment on those who break the rules

| Strongly disagree | Somewhat disagree | Neither agree nor disagree | Somewhat agree | Strongly Agree |
|---|---|---|---|---|
| ❏ | ❏ | ❏ | ❏ | ❏ |

❏ The healthiest society is the one based on competition between individuals

| Strongly disagree | Somewhat disagree | Neither agree nor disagree | Somewhat agree | Strongly Agree |
|---|---|---|---|---|
| ❏ | ❏ | ❏ | ❏ | ❏ |

❏ To deal with natural disasters individuals should prepare themselves before instead of waiting for someone to help them after

| Strongly disagree | Somewhat disagree | Neither agree nor disagree | Somewhat agree | Strongly Agree |
|---|---|---|---|---|
| ❏ | ❏ | ❏ | ❏ | ❏ |

❏ Many conflicts could be resolved by a fairer division of workload and natural resources

| Strongly disagree | Somewhat disagree | Neither agree nor disagree | Somewhat agree | Strongly Agree |
|---|---|---|---|---|
| ❏ | ❏ | ❏ | ❏ | ❏ |

❏ Natural disasters do not exist: it is only nature offended by man's arrogance and greediness, that rebels

| Strongly disagree | Somewhat disagree | Neither agree nor disagree | Somewhat agree | Strongly Agree |
|---|---|---|---|---|
| ❏ | ❏ | ❏ | ❏ | ❏ |

❏ When you do something for others you should not expect gratefulness

| Strongly disagree | Somewhat disagree | Neither agree nor disagree | Somewhat agree | Strongly Agree |
|---|---|---|---|---|
| ❏ | ❏ | ❏ | ❏ | ❏ |

DRAFT

❑ Natural disasters serve to remind us that making plans is useless: our lives are determined by forces that we cannot control

| Strongly disagree | Somewhat disagree | Neither agree nor disagree | Somewhat agree | Strongly Agree |
|---|---|---|---|---|
| ❑ | ❑ | ❑ | ❑ | ❑ |

**SECTION 6: MESSAGES AND CHANNELS TO SPREAD TSUNAMI ALERT**

25) **In your opinion, is it possible to spread an early warning before the arrival of a tsunami on the way to the coast?**
❑ Yes     ❑ No     ❑ I don't know (If answer is "not" or "I don't know" go to 25)

26) **In your opinion, who should warn population about the impending arrival of a tsunami on the way to the coast?**
1. _______________________________________________________________
2. _______________________________________________________________
3. _______________________________________________________________

27) **If a tsunami would hit your region's coastline, which channels would you prefer to receive an early warning?** (max three answers)

| |
|---|
| ❑ Television |
| ❑ Radio |
| ❑ Internet and social media |
| ❑ E-mail |
| ❑ Phone call (mobile or fixed telephone) |
| ❑ Be advised by friends, neighbours or relatives |
| ❑ SMS |
| ❑ Smarthpone application from official sources |
| ❑ Sirens or other acoustic devices |

---

## Author Response (AR1)

**Tsunami risk perception in Southern Italy: first evidence from a sample survey**

**by Andrea Cerase et al.**

1) **Comments from referees/public,**

*Anonymous Referee #1*

- The research presents a gap in the literature regarding the risk perception of citizens who did not recently experience a tsunami or those who think that an event like this will never occur in the Mediterranean ocean. I believe this is an interesting topic and it has scientific significance. Nevertheless, there are major issues in the manuscript that the authors must address to be suitable for publication.
- State three objectives is a big risk. According to the results of the article, the authors only address the first objective. The authors must delimit the scope of the paper. The three goals probably can be turned into three different papers.
- The hypotheses are not clear. The first RH1 must be redefined as a hypothesis and not as an affirmation. The RH2 is not relevant. Many studies already discover these differences.
- The methods and techniques section is deficient. A description of the study area and sample characteristics are missing. Also, there is not a description of the questionnaire, and most importantly, there is no evidence regarding the questions, neither the papers that were used to select the questions.
- The authors perform a focus group to test the questionnaire? Which was the no response rate?
- There are many errors about the numbers of the figures and tables, and many of them were not used in the text, such as Figure1, Table 1 and Figure 6.
- The Discussion section must be stated as "Results and Discussion" because it is confusing to the reader a Discussion section with so many results in it.
- Finally, I really think that the paper has significant relevance for the area, but the authors must rewrite the manuscript and organize it according to the journal standards.

*Anonymous Referee #2*

- This paper is relevant and addresses an area where there is a gap of knowledge, in the Mediterranean and other places in the world, especially where tsunami are infrequent, but could also be of high impact. I feel this paper provides the state of awareness of tsunamis in the region under consideration that would be helpful for implementing disaster risk reduction strategies.
- The questions used in the survey should be included as a supplement.
- The section on Research Hypothesis with the two Research Hypothesis needs to be rewritten and stated more clearly.

- There is a reference to Mitigation measures, but the paper does not address the state of mitigation (preparedness) efforts in the region.
- In the interpretation of the findings, there is no reference to preparedness and education outreach activities that have been carried out and may lead to a different risk perception, in addition to the presence of the volcanoes.
- Need to fix numbering of the Figures and verify reference to them in the text.
- The map of with distribution of interviewees, needs to have clearly labeled the places referred to in the text. It would also be helpful to see on this or another map, areas that have been the source of have been impacted by previous events and are referred to in the text.
- I was very confused by what was lumped together under "other broadcast media" in figure 2 and Table 3 (which the first column is not added up correctly) - it does not match the narrative. Did the question on INTERNET, also include Social Media specifically?
- For many Internet (web site) is very different from social media. I am suprised to not see a Social Media category.
- In 5.2 and Figure 3 it is not clear to me the interpretation of neutral? In the text it says for this category respondent "had no idea about its probability", is this really the case, it seems to be that the intermediate between Quite Likely and Unlikely would be "likely" or does neutral mean "I dont know".
- This is important because it affects the conclusion with regards to the state of perception.
- The Conclusion section needs to be rewritten and be more substantial with a focus on the findings from the survey. The authors go off on tangents, that are not related or a product of the surveys.
- One of the areas for further development would seem to be to integrate the perception of the tourists, which account for a greater number than residents.
- It refers to focus groups as non standard, it was my impression (I am not a social scientist) that these were valid. Many of the social science studies our agency supports use focus groups. What is meant by "collection of biographies".
- The English language is not of good quality in many sections, especially the Abstract,

**2) Author's response**

Dear Editor,
Dear reviewers,

Please find enclosed the revised version of the paper.

We would catch the opportunity to thank you and both two reviewers for constructive comments, as they helped us to improve the quality of our manuscript.
We have followed all the suggestions of both reviewers, as previously indicated in the point-to-point answers provided on July 22th, 2019.

In particular:
a.  we better delimited the scopes of the paper, as suggested by referee #1
b.  we have reorganised research hypotheses, as suggested;

c.  we have attached the Questionnaire in both Italian and English language as Supplementary Material;

d.  we have expanded the discussion on methodological and technical issues, providing more details on study area, data gathering methods, sampling features, response rates and possible limitations;

e.  we improved questionnaire description and the discussion of the literature that has been considered while drafting sections and questions;

f.  we have also provided a short note on the way two focus groups were used to involve scientists and laypeople in questionnaire development and testing;

g.  we have tried to provide a broader description of past risk communication and mitigation initiatives on tsunamis in Italy;

h.  we also tried to clarify some issues concerning the combination of different sources of knowledge and to discuss the related implications;

i.  we have thoroughly revised the Abstract and the Conclusion sections, according to the referees' suggestions, also including some additional results' description;

j.  we have improved the English language throughout the paper;

k.  we have added observations of past tsunamis in the map of Fig. 2, also improving its readability and hopefully its ability to clarify the historical reasons which can help to explain evidence.

l.  we have modified the text according to all the other reviewers' suggestions, in some cases modifying substantially some paragraphs.

We are definitely grateful to referee #1 because his/her attention to the aspects regarding methods and techniques, which helped us to highlight strengths and weaknesses of the research.

We are particularly grateful to referee #2 for one request, because it allowed us to realize a new, interesting result, notably the strong difference in risk perception between Calabria and Apulia, that we propose to interpret as due to the longer time elapsed since the last tsunami in the latter region.

We believe the revised version is substantially improved, and we remain however available for further clarifications.

The Authors:

Andrea Cerase

Massimo Crescimbene

Federica La Longa

Alessandro Amato.

3. **Author's changes in manuscript.**

[revised manuscript text omitted]

---

## Referee Report (RR1)

[referee-annotated manuscript omitted]

---

## Author Response (AR2)

Authors are grateful to both two reviewers and appreciate their valuable suggestions, which have led to a significant improvement of the paper. Authors provided all the suggested changes, both for text and figures.
* * *
**Report #1**
**Submitted on 27 Oct 2019**
**Anonymous Referee #3**
* * *
- Section 2 Question 2 : "Età" should be "Age."

- Section 3 Question 15: "Withou inlets" should be "Without inlets"
Spelling correction have been done as required.

- The manuscript gives the impression that the CAT-INGV is the only accredited Tsunami Service Provider (TSP) within NEAMTWS. It would have been appropriate to name other accredited TSPs of NEAMTWS (CENALT-France, NOA-Greece, and KOERI-Turkey), since CAT-INGV's service area has some overlaps with all of them.

Authors provided a better explanation of CAT-INGV and other operating roles and functions of TSPs within the NEAMTWS framework. See lines 105-109.

- Authors refer to two small tsunamis occurred in Dodecanese in 2017 due to earthquakes with magnitude 6.4 and 6.6. Presumably, these earthquakes are 12 June 2017 M 6.3 Plomarion (Lesbos, Greece) and 21 July 2017 M 6.6 Bodrum-Kos (Turkey, Greece) earthquakes, where the latter is mentioned by the authors in section 4.2. While it can be argued that the 21 July 2017 earthquake occurred in the vicinity of Dodeconese Islands, Plomarion is approximately 200km north with respect to Dodecanese Islands.

Authors agreed about the need to better specify the events to which they referred, and the places where they occurred. Slight corrections have been made to avoid misleading mention of seismic sources, which could be retrieved at lines 52-53.
* * *
**Report #2**
**Submitted on 29 Oct 2019**
**Referee #2: Christa von Hillebrandt-Andrade, christa.vonh@noaa.gov**
* * *
Referee #2 sent a hand-noted pdf file with a number of spelling suggestion to be made and some clarification request on relevant issues such as media as source of information (see notes at lines 451-456).
Any required correction has been implemented, along with the required clarifications. Authors also improved graphic quality of the mentioned figures, as required.

Track changes will exactly show where and how authors modified paper.

[revised manuscript text omitted]